# Ptch2/Gas1 and Ptch1/Boc differentially regulate Hedgehog signalling in murine primordial germ cell migration

Yeonjoo Kim [1], Jiyoung Lee[1], Maisa Seppala [2], Martyn T. Cobourne [2] & Soo-Hyun Kim [1✉]

Gas1 and Boc/Cdon act as co-receptors in the vertebrate Hedgehog signalling pathway, but the nature of their interaction with the primary Ptch1/2 receptors remains unclear. Here we demonstrate, using primordial germ cell migration in mouse as a developmental model, that specific hetero-complexes of Ptch2/Gas1 and Ptch1/Boc mediate the process of Smo de-repression with different kinetics, through distinct modes of Hedgehog ligand reception. Moreover, Ptch2-mediated Hedgehog signalling induces the phosphorylation of Creb and Src proteins in parallel to Gli induction, identifying a previously unknown Ptch2-specific signal pathway. We propose that although Ptch1 and Ptch2 functionally overlap in the sequestration of Smo, the spatiotemporal expression of Boc and Gas1 may determine the outcome of Hedgehog signalling through compartmentalisation and modulation of Smo-downstream signalling. Our study identifies the existence of a divergent Hedgehog signal pathway mediated by Ptch2 and provides a mechanism for differential interpretation of Hedgehog signalling in the germ cell niche.

[1] Molecular and Clinical Sciences Research Institute, St. George's, University of London, Cranmer Terrace, London SW17 0RE, UK. [2] Centre for Craniofacial and Regenerative Biology, King's College London, Guy's Hospital, London SE1 9RT, UK. ✉email: skim@sgul.ac.uk

Primordial germ cells (PGCs), the progenitors of gametes, provide a good model to study directional migration during development. The pluripotent PGCs rely upon precise communication by both secreted morphogens acting at long distance and local signalling produced by the immediate micro-environment[1–4]. In mouse, PGCs arise from the posterior extra-embryonic mesoderm around embryonic stage (E)7.5 and migrate into the definitive endoderm, continuing through the developing hindgut (HG) until E9.5. Around E10.5, PGCs directionally migrate from the dorsal body wall, finally arriving at the bilateral genital ridges (GR) between E10.5 and 11.5, where they become immotile and differentiate into either spermatozoa or oocytes, integrating with the developing somatic gonads[2,3,5]. Studies in *Drosophila* and zebrafish suggest that Hedgehog (Hh) signalling is involved in the development of PGCs, but does not function as a fate determinant or guidance molecule[6–8]. The role of Hh signalling in mouse PGCs still remains ambiguous, although the aorta, gonad, mesonephros and GR are suggested to be the main source of chemo-attractantion[3,9,10].

The role of Hh in chemotaxis has been demonstrated in different developmental contexts[11–14]. Binding of Hh to the two paralogue Patched (Ptch1/2) receptors releases the Smoothened (Smo) G-protein coupled receptor, which allows its translocation to the primary cilia and the de-repression of Smo-dependent signalling. Ptch2 shares structural similarities with Ptch1, including extracellular ligand-binding loops and transmembrane domains, but has much shorter intracellular amino- and carboxy-terminals[15–17]. Like Ptch1, Ptch2 interacts with all mammalian Hh ligands (Sonic hedgehog, Shh; Desert hedgehog, Dhh; Indian hedgehog, Ihh) with a similar affinity and forms a complex with Smo. Contradicting reports claim that Ptch2 possesses either similar, weaker or no repressive activity on Smo during Hh signalling, compared with Ptch1[16,18–20]. The embryonic expression pattern of *Ptch2* is distinct from *Ptch1*, mainly in the skin and testis, and overlapping with *Shh*[18]. *Ptch2*-hypomorphic mice are viable and fertile, whilst *Ptch1* mutants are embryonic lethal, suggesting that Ptch1 is the major regulator of Hh signalling and Ptch2 has a redundant function compensatable by Ptch1[21–23]. Cells lacking Ptch1 remain sensitive to Hh in a chemotaxis assay, and Ptch2 can mediate Hh-induced motile responses in the absence of Ptch1[24].

The Gli family of transcription factors mediate the canonical Hh pathway to regulate cell fate and patterning in accordance with the ligand gradient. Gli-independent non-canonical signalling also occurs, which does not require Smo localisation to the primary cilia. It is thought that non-canonical Hh signalling regulates cytoskeletal rearrangement in differentiated cells after specification[25,26], endothelial cell migration in pro-angiogenic responses[27] and axon guidance through the activation of Src kinases[28]. A timely switch from canonical to non-canonical signalling can also occur during normal development[29]. Notably, Smo proteins located outside the cilia are believed to be responsible for non-canonical chemotactic responses, suggesting that Smo localisation might be the critical determinant in the selective engagement of canonical versus non-canonical pathways[25,30]. Since the majority of Smo proteins reside outside of the primary cilia, it has also been speculated that non-canonical signalling may represent a more general and robust Hh response in the normal state[26].

A number of obligatory co-receptors for Hh have been identified in vertebrates, which include Boc (bioregional Cdon-binding protein), Cdon (cell-adhesion-molecule-related/down-regulated by oncogenes, also called as Cdo) and Gas1 (Growth arrest-specific gene 1). Cdon and Boc belong to a family of cell adhesion proteins[31], while Gas1 is structurally distinct and localises at plasma membrane rafts via a glycosylphosphatidylinositol (GPI) anchor[32]. These co-receptors can bind Hh ligand independently of Ptch1 and facilitate ligand–receptor interaction at the cell surface[33,34]. Boc and Cdon have partially redundant and distinct tissue-specific roles in Hh regulation during myogenic differentiation and axon guidance[35–38]. Gas1 positively regulates Shh signalling in the neural tube and forebrain[39,40] but may exert a negative effect in the somite and mandibular arch at high concentrations[41–43]. Certainly, the developmental defects observed in *Gas1* mutant mice suggest a complex relationship between Gas1 and Shh[44,45]. Since the positive effects of Gas1 are evident in areas of low Hh concentration, it has been speculated that Gas1 may extend the range of ligand gradient, translating a low peripheral chemical concentration into cellular activity[46]. In contrast, Boc/Cdon are implicated in areas of high Hh concentration, potentially required for the maximum canonical response[47].

A key question is how Ptch1 and Ptch2 recognise and respond differentially to Hh ligand in the production of different cellular outcomes, particularly when considering that both canonical and non-canonical processes require Smo. Moreover, the role of co-receptor function in Ptch2-dependent signalling is completely unknown. Here we reveal that PGCs are naturally unciliated but remain sensitive to locally secreted Hh ligand through an exclusive interaction between Ptch2 and Gas1, while ciliated somatic cells immediately surrounding PGCs rely on interaction between Ptch1 and Boc. The molecular mechanisms leading to the de-repression of Smo after Hh ligand reception are different in Ptch1/Boc and Ptch2/Gas1-mediated signalling events, resulting in distinct downstream signal pathways acting in parallel to Gli-dependent transcription. Our findings provide a previously unreported mechanism for Ptch2-specific signal pathways involving the cAMP-responsive element binding protein (Creb) and Src tyrosine kinase. We propose that these distinct molecular mechanisms, mediated by different Ptch receptors and their co-receptors, may be a key determinant in the selective activation of distinct downstream signalling pathways. This may provide the required specificity and fine-tuning of Hh signalling, leading to either cell fate specification or post-specification cellular behaviour, such as motility.

## Results

**Expression of Shh, Gas1 and Boc in PGC migration niche.** To evaluate the involvement of Hh signalling during the migration of post-specification PGCs in mouse, we examined the spatio-temporal expression of Hh pathway genes in the gonadal anlage of E9.5–11.5 embryos. We found that *Ptch1*, *Gli1/2/3*, *Dhh* and *Shh*, but not *Ihh*, were expressed along with a germ cell marker *Stella* in the HG at E9.5 and in the GR at E10.5–11.5, where PGCs localise (Fig. 1a). The mRNAs of *Boc* and *Gas1* were detected in the urogenital ridges but *Cdon* was not, indicating specific involvement (Fig. 1b). Further examination of the expression patterns of *Shh*, *Gas1*, *Boc* and *Cdon* by in situ hybridisation and immunostaining of transverse sections showed that at E9.5 when post-specification PGCs are motile but still restricted within the HG, *Shh* and *Gas1* were expressed in overlapping patterns along the HG spreading dorsally towards the urogenital ridges, while *Boc* showed a broad signal which was much less intense compared with *Shh* or *Gas1* (Fig. 1c, top panel). As development continued into E10.5 when PGCs rapidly exit from the HG migrating towards the GR, the *Gas1* signal seemed to follow the PGC migration route, moving dorsally from HG towards the mesentery and GR, while *Boc* expression was broadly present in these regions increasingly visible throughout the developing GR (Fig. 1c, bottom panel). Detailed examination after co-staining with the PGC marker SSEA1 revealed that Gas1 signal exclusively

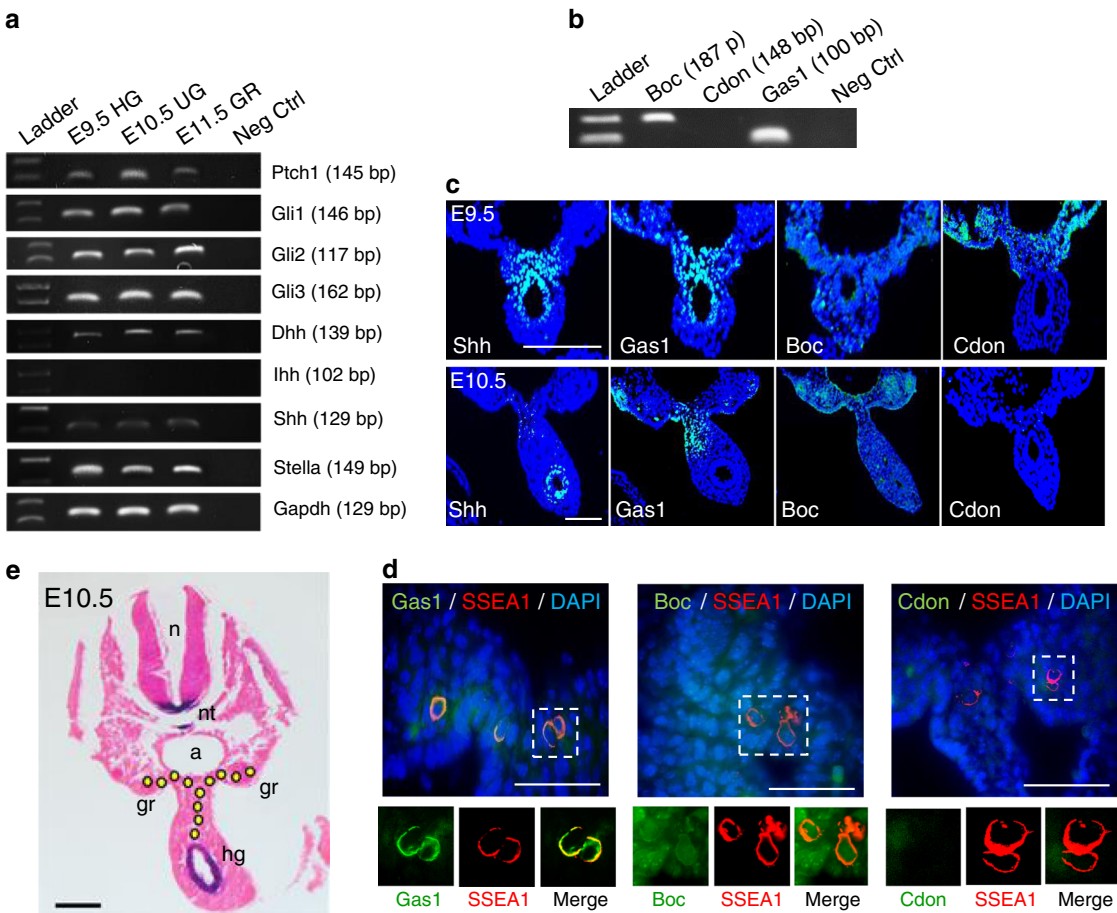

**Fig. 1 Hh pathway genes and co-receptors are expressed in the PGC migratory niche in the mouse. a** RT-PCR analyses of Hh pathway genes in E9.5 hindgut (HG), E10.5 urogenital ridge (UG) and E11.5 genital ridge (GR). **b** RT-PCR analysis of Ptch co-receptors, *Boc*, *Cdon* and *Gas1* in E10.5 UG. **c** Immunofluorescence staining of Shh, Gas1, Boc and Cdon on E9.5 and E10.5 embryo sections. Scale bar, 50 μm. **d** Co-staining of germ cell marker SSEA1 and Gas1, Boc or Cdon on the genital ridge area of E10.5 embryos. Scale bar, 50 μm. **e** In situ hybridisation of *Shh* on the transverse section of E10.5 embryo counterstained with haematoxylin. The PGC migratory route from HG heading towards UG is illustrated by circled dots. n neural tube, nt notochord, a aorta, gr genital ridge, hg hindgut. Scale bar 100 μm.

overlaps with SSEA1 but Boc signal is ubiquitous (Fig. 1d), thus migrating PGCs specifically express Gas1, while Boc is broadly expressed in the PGCs and the surrounding mesenchyme. In contrast, Cdon is not detected in PGCs or the adjacent migration niche during this period (Fig. 1c, d). Interestingly, *Shh* expression remained restricted within the HG during this period (Fig. 1c, e). The fact that PGCs migrate away from the source of high *Shh* expression confirms that Shh is not a chemoattractant for PGCs.

**Shh enhances PGC motility without affecting directionality.** To further define the role of Shh in PGC migration, we conducted a chemotaxis assay by co-culturing PGCs juxtaposed to 293 cells transfected with either human full-length *SHH* construct or an empty vector (Fig. 2a, b). After 24 h incubation, we found that similar numbers of PGCs migrated towards the 293 cells regardless of Shh secretion (Fig. 2c, right panel), confirming that Shh is not a chemoattractant for PGCs, as suggested from the embryo expression data (see Fig. 1). However, when we compared the migration distance, PGCs located among the Shh-expressing 293 cells migrated significantly further from the middle line (Fig. 2c, left panel), implying that Shh may enhance the motility or speed of migration. To test this notion, we generated time-lapse movies of embryo slice cultures derived from the *Stella*^GFP transgenic mice, which allowed us to monitor the migratory

behaviours of GFP-expressing PGCs in live embryos (Supplementary Fig. 1 and Supplementary Movies 1–4). To assess the effects of Hh signalling, we treated the slice culture medium with the Hh agonist (purmorphamine) or antagonist (cyclopamine), respectively, during the 10 h of live imaging (Fig. 3a). Analyses of the velocity, accumulated distance (total cell path travelled), Euclidean distance (the shortest distance between the start and endpoints) and directionality (the ratio between Euclidean distance and accumulated distance) revealed that purmorphamine enhanced the speed and distance of migration, while cyclopamine significantly reduced these values, compared with the solvent control. Consistently, tomatidine, a drug with a similar structure to cyclopamine but lacking any inhibitory activity, had no effects on PGC migration within these parameters (Fig. 3b). Notably, none of the treatments altered the directionality of migration (Fig. 3a, b), signifying that Hh enhances the motility of PGCs without influencing directionality. PGC survival time was similar in all treatment groups (Fig. 3c), indicating no general cytotoxicity from these treatments. Since the embryo size expansion over the period of live imaging was similar in all treatment groups (Fig. 3d, Supplementary Fig. 3), it is unlikely that the altered PGC migration was simply a consequence of morphological changes in the surrounding tissues of the growing embryo. To directly demonstrate the effects of Hh signalling on the intrinsic motile capacity of PGCs, we generated primary cultures of dissected GR,

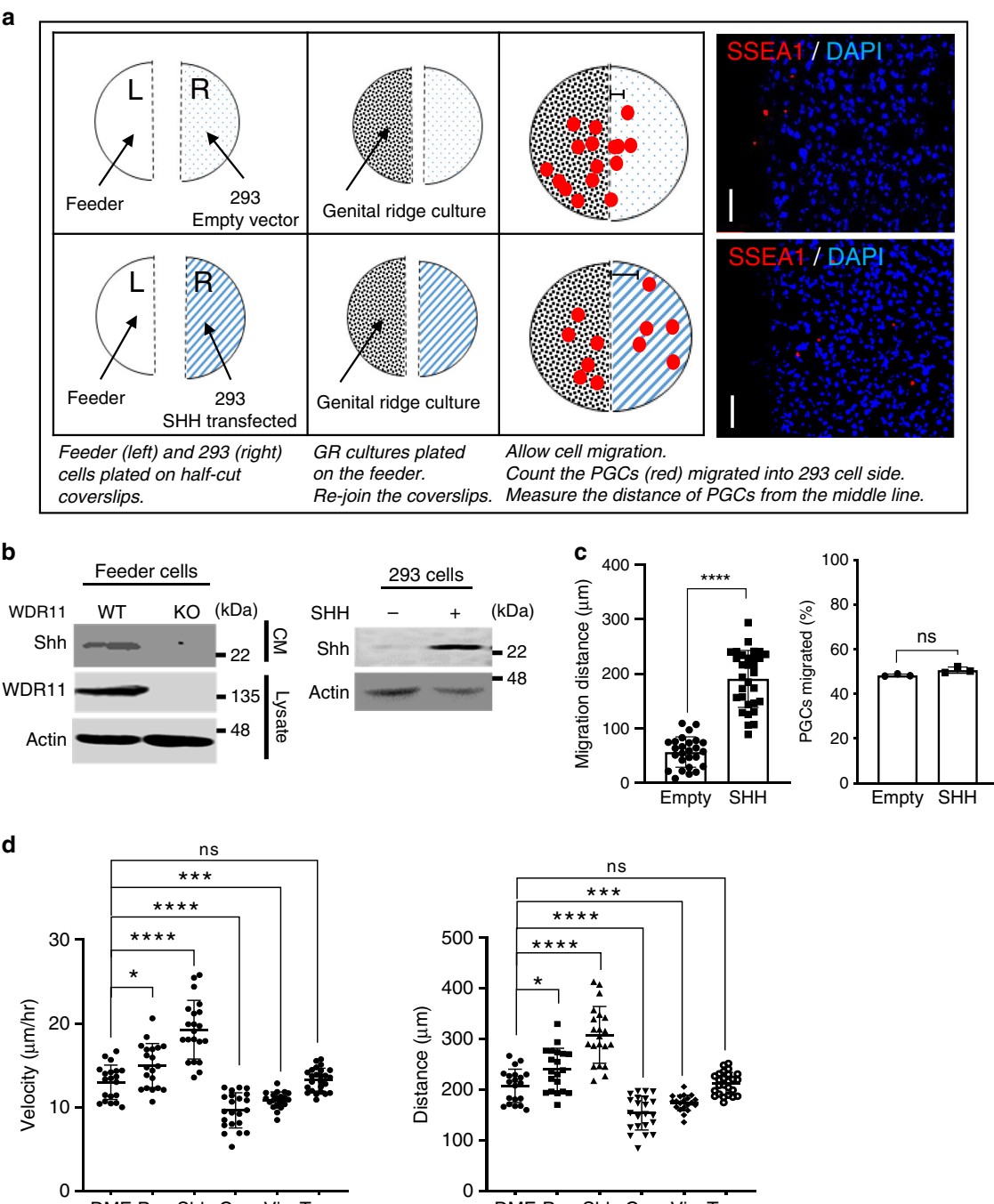

**Fig. 2 Hh enhances PGC migration without chemoattraction. a** Schematic representation of the chemotaxis assay. NIH3T3 cells deleted for the endogenous *Wdr11* by CRISPR/Cas9, and therefore unable to secrete Shh, were used as the feeder. On average, ~30,000 cells of primary GR culture derived from five embryos at E10.5 were plated on top of the feeder. For the source of Shh secretion, 293 cells transfected with a human full-length *SHH* expression construct were plated in comparison to the empty vector. After 24 h of incubation, cells on both sides of the coverslips were analysed by immunofluorescence. Scale bar, 50 μm. **b** Western blot analyses validated *Wdr11* KO feeder lacking Shh secretion in the conditioned medium (CM), while the transfected 293 cells producing Shh. B-actin is a loading control. Representative blots and cell images from three independent experiments are shown. **c** Total numbers of PGCs (SSEA+) and their distance from the middle (the dotted line in **a**) were assessed from ten random fields. Percentages of PGCs migrated towards the transfected 293 cells were similar in both groups, but the average distance of migration was significantly higher in *SHH*-expressing cells (*n* = 35) compared with empty vector (*n* = 30). Data are presented as mean ± SD from three independent experiments (****P < 0.0001; NS, P > 0.05). **d** Intrinsic random motility of PGC was assessed by live time-lapse imaging (see Supplementary Movies 5–10). The average of velocity and accumulated moving distance of GFP-positive cells in random fields of view tracked for 16 h in three biologically independent experiments are shown. Data from solvent dimethyl formamide (DMF, *n* = 20), purmorphamine (Pur, *n* = 20), cyclopamine (Cyc, *n* = 22), visomodegib (Vis, *n* = 20) and tomatidine (Tom, *n* = 25) treatment groups are analysed using unpaired *t*-test with Welch's correction (*P = 0.0111; ***P = 0.0003; ****P < 0.0001). Source data are provided as a Source Data file.

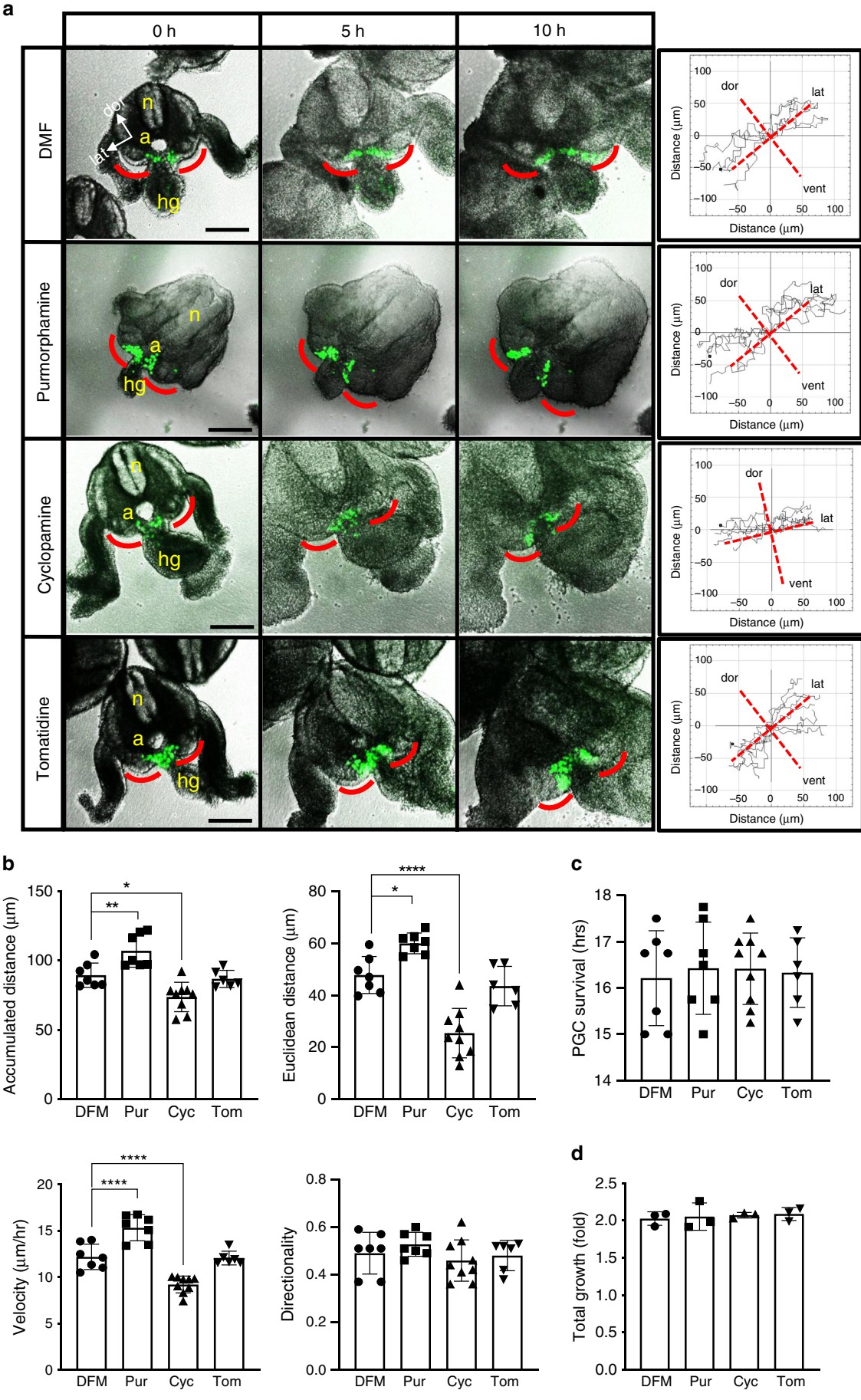

**Fig. 3 Hh signal regulates the motility of PGC but not the directionality. a** Representative images of E10.5 *Stella*GFP embryo slice cultures. Frames at $t = 0$, 5 and 10 h from the live imaging (Supplementary Movies 1–4) after treatment with the solvent (DMF, $n = 7$), 40 μM purmorphamine (Pur, $n = 7$), 40 μM cyclopamine (Cyc, $n = 9$) or 40 μM tomatidine (Tom, $n = 6$) in biologically independent experiments. The GFP-positive PGCs migrating towards the genital ridges (red lines in the embryo image) were tracked using ImageJ plugin. Trajectories of PGCs were generated by placing the starting point of the individual PGC track onto the same point. The direction of migration in relevance to the embryo orientation is shown. n neural tube, a aorta, hg hindgut, dor dorsal, lat lateral, vent ventral. Scale bar, 100 μm. **b** The effects of Hh agonist/antagonists on the migratory behaviour of PGCs in different parameters are assessed from 7 to 10 PGCs in one embryo slice. Data from DMF ($n = 7$), Pur ($n = 7$), Cyc ($n = 9$) and Tom ($n = 6$) treatment groups are presented as mean ± SEM. One-way ANOVA followed by Dunnett's test (*$P < 0.05$; **$P < 0.01$; ****$P < 0.0001$). **c** Cell survival was assessed by the number of hours that the fluorescence signal from GFP was detected during the live imaging of slice cultures shown in **b**. One-way ANOVA indicated that cell survival was not different among treatment groups ($P = 0.9651$). **d** Growth rate of each embryo, presented as the fold expansion of the embryo size during the live imaging, was assessed by measuring the total trunk area of each embryo slice at the beginning and end of the live imaging ($t = 0$ h and $t = 10$ h) using ImageJ. Error bars represent SD from three biologically independent experiments. One-way ANOVA indicated that growth rate was not affected by different treatments ($P = 0.9186$). Source data are provided as a Source Data file.

where the PGCs and their neighbouring somatic cells were dispersed in single-cell suspensions and cultured as a monolayer of cells in a dish. When we analysed the random motility of isolated PGCs by time-lapse imaging (Supplementary Movies 5–10), treatment of Hh agonists (Purmorphamine and recombinant Shh-N protein) increased the random motility of PGCs in terms of velocity and accumulated distance. Notably, treatment of Hh antagonists (cyclopamine and vismodegib) inhibited the random motility compared with the solvent or tomatidine (Fig. 2d). We did not find evidence of general toxicity from the drug treatments at the concentrations we used (Supplementary Fig. 2), although lower concentrations of vismodegib and cyclopamine had been used in some explant cultures[48]. Combined, these data suggest that Hh signalling directly regulates the intrinsic motile capacity of PGCs.

**Naturally unciliated PGCs remain responsive to Hh signal**. Our live imaging data demonstrated that PGCs migrate as single cells dispersed throughout the migratory niche mesenchyme, rather than a collective migration as a group or long line (see Supplementary Movies 1–4). We, therefore, speculate that the motility of PGCs primarily relies on the communication of individual PGCs with their immediate surroundings. To investigate how the Hh signal pathway operates in PGCs and soma, we examined their signalling responses in primary cultures of GR. After treatment with purmorphamine for 18 h, the somatic cells (negative for SSEA1 or Stella), showed a distinctive accumulation of Smo and Gli3 to the primary cilia, as expected from ciliated cells upon activation of the canonical Hh signalling pathway. However, PGCs showed significantly increased levels of Smo and Gli3 proteins within the cytoplasm (Fig. 4a, b). Since PGCs exhibited an unexpected pattern, we explored if there was any alteration of primary cilia in PGCs. Surprisingly, the cilia axoneme identifiable by Arl13b or acetylated tubulin staining was mostly absent in the PGCs, although the basal body was visible after gamma-tubulin or CEP164 staining (Fig. 4c, d). These results indicated that PGCs naturally lack primary cilia, but the Hh signal pathway can still be induced in these cells through direct activation of Smo and that Smo proteins localised outside the cilia can mediate Hh signalling, as previously observed from cilia-knockout cell lines[26].

**Migrating PGCs specifically express Ptch2 and Gas1**. We next asked whether Hh signalling is received differently in PGCs due to the lack of primary cilia. Immunofluorescence studies of E10.5 GR primary cultures revealed specific expression of Ptch2 on the PGCs, while Ptch1 was detected only on the somatic cells (Fig. 5a). Quantitative RT-PCR analyses on the sorted populations of PGCs (*Stella*+/GFP+) and the somatic cells (*Stella*-/GFP-) isolated from E10.5 *Stella*GFP embryos after FACS (Supplementary Fig. 4)

further confirmed the exclusive expression of *Ptch2* and *Ptch1* mRNA on PGCs and soma, respectively (Fig. 5b). We also confirmed in the primary GR cultures that Gas1 was exclusively present on PGCs, while Boc was ubiquitously detected in both cell types but Cdon was absent (Fig. 5a), consistent with our findings from embryo tissue sections (see Fig. 1d). Are these specific combinations of receptors functionally important for Hh-dependent signalling and PGC migration? To test this notion, we examined the phosphorylation status of Src, an indicator of cytoskeletal rearrangement during Hh-induced motility[28]. In E10.5 wildtype (WT) embryos, the highly motile PGCs were stained positive for p-Src. However, *Gas1* knockout (KO) mouse embryos showed a significant reduction in p-Src, even though Ptch2 and Boc expression remained similar (Fig. 5c, d), suggesting that Gas1 plays an important role in the motile capacity of PGCs, potentially mediating Ptch2-dependent Hh signalling, and Boc cannot compensate for the loss of Gas1. Since the somatic cells expressing Boc/Ptch1 did not show p-Src signal (Fig. 5c), Hh signalling in these cells may not lead to any significant focal adhesion kinase-mediated events, such as cell motility or survival. Combined, these data support a model where Hh ligand is received by Ptch2/Gas1 receptor complexes on PGCs, while Ptch1/Boc receive the ligand in somatic cells.

**Gas1-mediated Hh binding to Ptch2 causes Smo de-repression**. To clarify the dynamic interactions of these receptors, we performed co-immunoprecipitation (Co-IP) analyses at different time-points after Shh stimulation. We used NIH3T3 cells as a well-established model, which normally express Ptch1, Gas1, Boc and Cdon, but not Ptch2 (Supplementary Figs. 5 and 7). To study Ptch1/2 receptors individually without interference from each other, we removed the endogenous *Ptch1* by CRISPR/Cas9 (Supplementary Fig. 6, lower panel) and introduced a *Ptch2* overexpression construct. In the unstimulated state, the Ptch2 immuno-complex contained Smo without participation of Gas1 or Boc. After the addition of Shh for 40 min, Gas1 and Shh were recruited to Ptch2, which caused a complete dissociation of Smo from Ptch2 (Fig. 6a). A reciprocal Co-IP using antibody against endogenous Gas1 confirmed that Gas1 bound to Ptch2 only when Shh ligand was present. Gas1 did not interact with Smo either before or after Shh addition. Boc remained unbound to Ptch2 (Fig. 6a). The specificity of these interactions was further validated by a reciprocal Co-IP in empty-vector transfected cells, which did not show any non-specific binding (Fig. 6a, bottom panel). Based on these data, we hypothesise that Gas1 mediates Shh binding to Ptch2, and the formation of a Ptch2/Gas1/Shh tertiary complex has an immediate effect on the release of Smo from Ptch2 (Fig. 6b). If this is the case, Shh would not be able to bind to Ptch2 or cause Smo release in the absence of Gas1. When

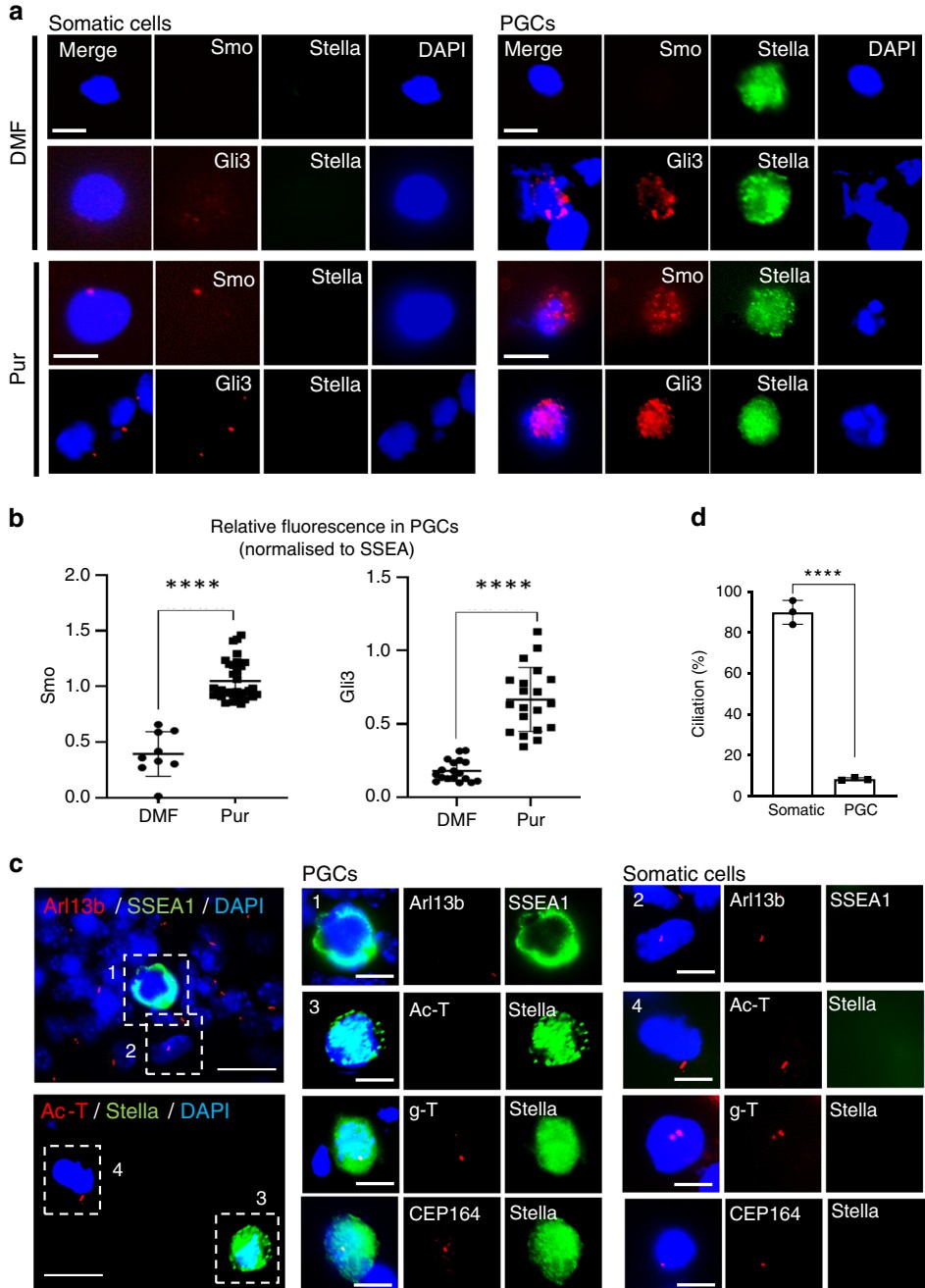

**Fig. 4 PGCs are naturally unciliated but still responsive to Hh signalling. a** Primary cultures of GR were treated with DMF or Pur for 18 h and analysed by immunofluorescence staining for Smo and Gli3. The positive and negative staining for Stella, a germ cell marker, distinguished PGCs and somatic cells, respectively. The merged images are shown without Stella signal for improved clarity. Representative images from three biologically independent experiments are shown. Scale bar, 10 μm. **b** Dot plots showing the relative fluorescence intensity values of Smo and Gli3 signal observed in PGCs treated with DMF or Pur, which were normalised to the fluorescence intensity values of Stella in each cell. Unpaired *t*-test with Welch's correction indicates a significant increase in Smo (DMF (*n* = 9), Pur (*n* = 31), *P* < 0.0001) and Gli3 (DMF (*n* = 18), Pur (*n* = 20), *P* < 0.0001). **c** Cilia staining of primary cultures of dissected E10.5 mouse GR tissues. Arl13b and acetylated tubulin are used for axoneme staining and gamma-tubulin and CEP164 are used for basal body staining. SSEA1 and Stella are used as germ cell markers. Representative images from three independent experiments are shown. Scale bar, 10 μm. **d** Ciliation frequency of the PGCs (*n* = 158) and the somatic cells (*n* = 971) observed from GR cultures. Error bars represent SD of three independent experiments. Each data point represents one experiment. Unpaired *t*-test, two-tailed (****\*\*\*\*P* < 0.0001). Source data are provided as a Source Data file.

we repeated the Ptch2 Co-IP experiment in *Gas1* KO NIH3T3/Cas9 cells (Supplementary Fig. 6, upper panel), Ptch2 was indeed unable to receive Shh ligand and Smo remained within the Ptch2 immuno-complex, even after the addition of Shh (Fig. 6c), supporting our model (Fig. 6d).

**Boc mediates slow and gradual release of Smo from Ptch1.** We next investigated Boc/Ptch1 interactions using similar approaches after *Ptch1* overexpression. Multiple reciprocal Co-IP experiments showed Boc already bound to Ptch1/Smo in the unstimulated state (Fig. 7a). After the addition of Shh for 40

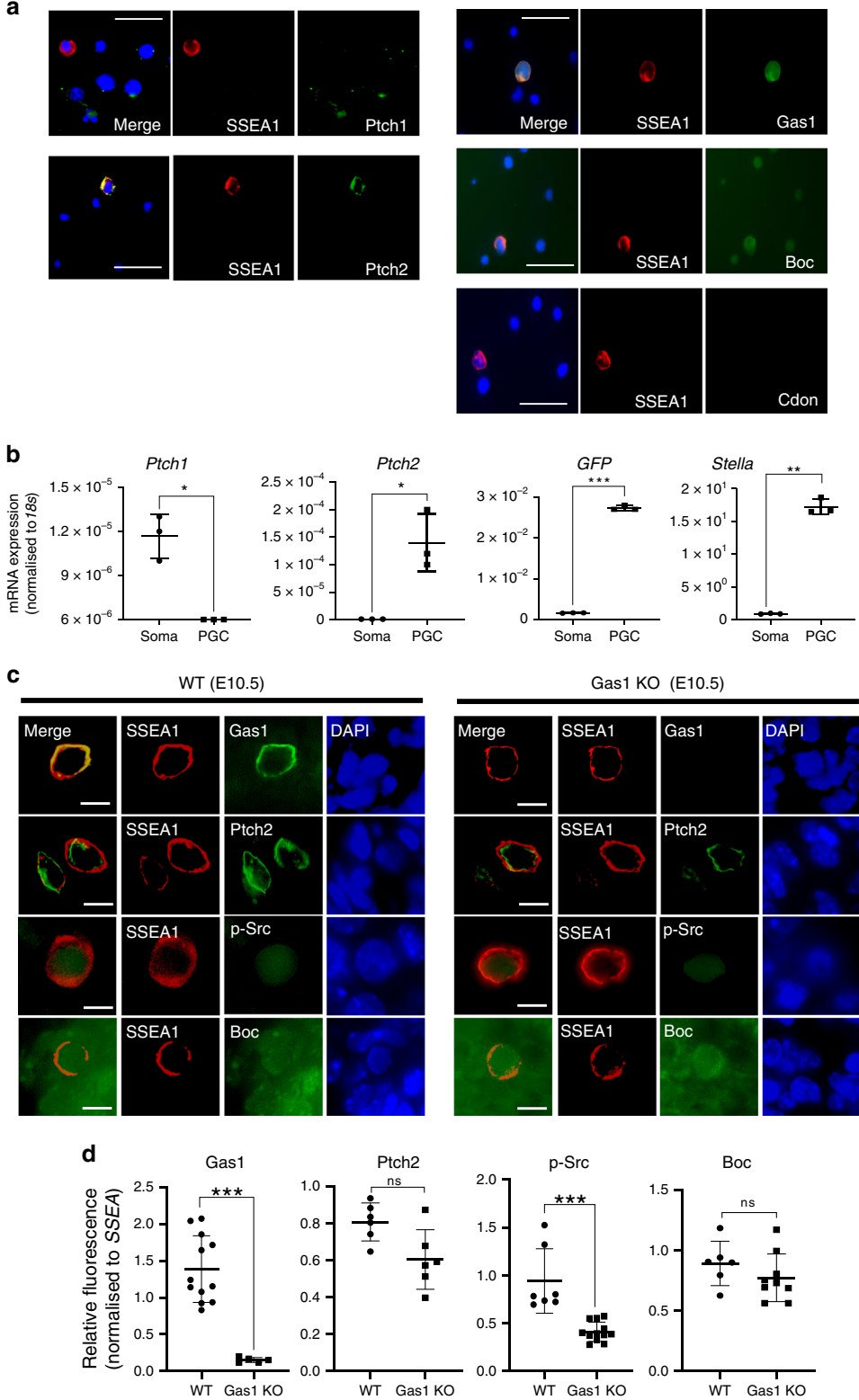

min, the Ptch1/Boc/Smo complex recruited Shh, which caused only a partial release of Boc and Smo from Ptch1. At 24 h of Shh treatment, we observed a complete release of Smo from Ptch1. Interestingly, Shh was increasingly bound to Ptch1 at this time, while Boc became completely dissociated from it (Fig. 7a), indicating that Boc transferred Shh to Ptch1 and then left. A reciprocal Co-IP of endogenous Boc confirmed that Boc was indeed complexed with Ptch1 and Smo before the addition

of Shh. At 40 min after ligand addition, the binding interactions of Boc with Smo and Ptch1 became weaker, while Boc/ Shh binding became more prominent. After 24 h of Shh treatment, Boc completely dissociated from Ptch1, Smo and Shh (Fig. 7a). Gas1 was not involved in the Ptch1 complex at any time-points, confirming the exclusive partnership. Combined, these results suggest a model where Boc/Ptch1-mediated de-repression of Smo occurs gradually, at least through two

**Fig 5 Ptch1/2 and co-receptors are differentially expressed in PGCs and somatic cells. a** Expression of Ptch1, Ptch2, Gas1, Boc and Cdon demonstrated by immunofluorescence analyses of primary cultures of dissected E10.5 mouse GR, with co-staining of SSEA1. Representative images from three independent experiments are shown. Scale bar, 50 μm. **b** Single-cell suspensions from freshly-dissected GR of E10.5 GFP^Stella embryos were sorted by FACS (Supplementary Fig. 4) and the expression of *Ptch1, Ptch2, GFP* and *Stella* in GFP-positive (PGCs) and GFP-negative (somatic cells) populations analysed by quantitative RT-PCR. Each data point represents one experiment. Values normalised to *18s* rRNA were plotted with error bars representing SD ($n = 3$) after unpaired *t*-test (*$P < 0.05$, **$P < 0.005$, ***$P < 0.001$). **c** Immunofluorescence analyses of Gas1, Ptch2, p-Src, Boc and SSEA1 on the GR tissue sections of WT and Gas1 KO embryos at E10.5. The merged images are shown without DAPI signal for improved clarity. Representative images from three independent experiments are shown. Scale bar, 10 μm. **d** Dot plots showing the relative fluorescence intensity values of Gas1, Ptch2, p-Src, Boc signal observed in PGCs which were normalised with the fluorescence intensity values of SSEA in each cell. Gas1 KO embryos showed a significant decrease in signal intensity in Gas1 (WT ($n = 12$), KO ($n = 5$), ****$P < 0.0001$) and p-Src (WT ($n = 7$), KO ($n = 12$), **$P = 0.0054$) but no significant changes in Ptch2 (WT ($n = 6$), KO ($n = 6$), $P = 0.0564$) and Boc (WT ($n = 6$), KO ($n = 9$), $P = 0.2583$). Error bars represent SD after unpaired *t*-test with Welch's correction. Source data are provided as a Source Data file.

steps after Hh ligand reception by Boc, which transfers the ligand to Ptch1 (Fig. 7b).

**Ptch2 and Ptch1 elicit differential downstream signalling.** These data suggested that Gas1 is a preferential co-receptor that is likely to facilitate Ptch2-dependent Hh signalling through a rapid Smo de-repression process. In contrast, the release of Smo from the Ptch1/Boc receptor complex occurs gradually over an extended period of time, conforming to the well-established timeframe of canonical Hh signalling. Therefore, we next compared the dynamics of downstream signalling responses mediated by Ptch1 and Ptch2. First, we examined the induction of *Gli1* and *Gli3* mRNA as a well-established readout of canonical Hh signalling. Transcription of *Gli1* and *Gli3* in *Ptch1*-expressing cells gradually increased after Shh addition, reaching to a maximum (7.4 fold and 5.1 fold, respectively) at 24 h. In contrast, in *Ptch2*-expressing cells, the induction of *Gli1/3* remained at minimal levels even after 24 h (1.8 fold and 1.6 fold, respectively) (Fig. 8a), agreeing with the view that Ptch2 is less efficient than Ptch1 in mediating canonical Hh signalling.

It is widely accepted that Hh-dependent induction of cAMP levels and protein kinase A (PKA) activity at the cilium base regulate the coupling of Smo with inhibitory GTP-binding proteins (Gαi)[49,50]. However, this compartmentalised local change of cAMP is undetectable in whole-cell extracts[51]. Since cAMP-PKA signalling activates Creb in NIH3T3 cells[52], we assessed the level of phospho-Creb by Western blot at different time-points after Shh treatment. *Ptch1*-expressing cells did not show any p-Creb, agreeing with previous findings that Hh alone does not cause global p-Creb induction[51]. Surprisingly, *Ptch2*-expressing cells did show an induction of p-Creb at 40 min, which further increased at 24 h (Fig. 8b). Therefore, the de-repression of Smo within the context of Ptch2 has different effects to Ptch1, eliciting Creb-dependent global responses. Since this phenomenon occurred without any exogenous manipulations for cAMP, adenylyl cyclase or PKA, it provides clear evidence for the existence of a p-Creb-dependent gene regulatory network directly downstream of Hh/Smo, operating parallel to Gli transcription factors.

The tyrosine kinase Src acts as a key regulator of cell motility by phosphorylating multiple protein substrates that control the cytoskeleton and focal adhesion. Since migrating PGCs show positive staining for p-Src (see Fig. 5c), we investigated if Ptch2 is involved in this signalling pathway. The level of p-Src was undetectable in *Ptch1*-expressing cells, but *Ptch2*-expressing cells displayed a substantial level of p-Src, which further increased upon Hh stimulation (Fig. 8b). Combined, these data demonstrate that Ptch2-dependent Shh signalling leads to global changes in cellular p-Creb and p-Src, that are distinct from Ptch1-mediated signal responses.

**Essential requirement of Gas1 in Ptch2-specific signalling.** Co-IP data demonstrated that Gas1 does not interact with Ptch1 although Shh remains bound to Gas1 (Fig. 7a), thus Gas1/Shh explicitly interacts with Ptch2 even in the presence of Ptch1 and does not compete with Boc/Shh for Ptch1 binding. If so, a Gas1 deficiency should specifically interfere with Ptch2-dependent signalling without affecting Ptch1-mediated responses. Our analyses in *Gas1* KO NIH3T3/Cas9 cells confirmed that Ptch2-dependent induction of *Gli1/3* was completely abolished by the lack of Gas1, while those of Ptch1-dependent responses were virtually unaffected (Fig. 8a). Likewise, the induction of p-Creb and p-Src in Ptch2-expressing cells was completely lost, even in the presence of endogenous Boc and Cdon (Fig. 8b). Since Gas1 mutant cells express normal primary cilia (Supplementary Fig. 8), the observed disruptions of Hh-induced responses were not due to defective cilia. Together, these data support that Gas1 is essential for Ptch2-dependent Hh signalling, while Boc/Ptch1 is likely to function as the main receptor for canonical signalling, as previously suggested[47].

**Shh induces Ptch2-specific signal responses in PGCs, but not in the soma.** To further validate these findings, we examined signal responses in primary cultures of GR expressing these receptors endogenously (Fig. 5). If our model is correct, a brief treatment with Shh ligand should induce Ptch2/Gas1-specific responses in PGCs, but not in somatic cells. Immunofluorescence results showed the phosphorylation of Creb and Src in PGCs even after 10-min treatment with Shh ligand, but not in the surrounding somatic cells (Fig. 9a, b). These data support the notion that post-specification PGCs are likely to respond rapidly to locally secreted Hh ligand via Gas1/Ptch2, eliciting a Smo-downstream signalling pathway involving p-Creb and p-Src, which does not occur in non-migratory mesenchymal cells expressing Ptch1/Boc.

**Gas1-deficient mice show ectopic PGCs and subfertility.** To confirm the essential role of Gas1 for the migration of PGCs in vivo, we examined the establishment of PGCs and fertility phenotypes in *Gas1* KO mice. *Gas1*-null embryos contained significantly fewer numbers of PGCs arriving in the GR at E10.5 with an increased presence of ectopic PGCs in the mesentery and HG areas (Fig. 10a, b), indicating a delayed migration compared with WT. Since the total numbers of SSEA1-positive PGCs remained similar, Gas1 deficiency did not affect the specification and survival of PGCs (Fig. 10b). However, the reduced number of PGCs in the developing GR would be expected to result in reduced numbers of germ cells in the gonads at birth. We investigated the fertility of *Gas1* mutants by comparing the average litter sizes after timed-mating, and found *Gas1*-null females produced significantly fewer litters compared with the heterozygotes, indicating a subfertility (Fig. 10c).

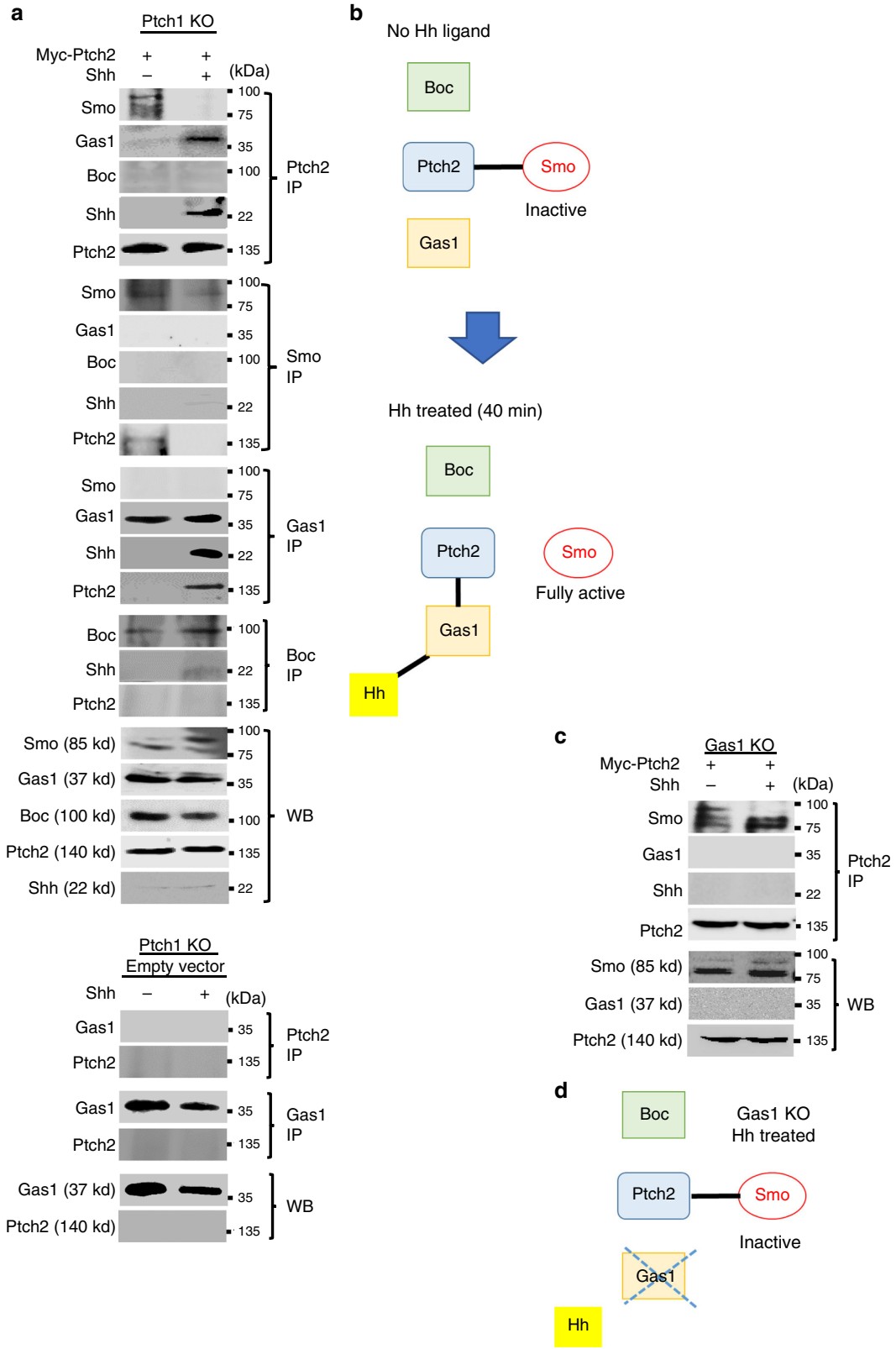

## Discussion

Although the predominant model is Shh acting as a morphogen to induce a Ptch1-dependent signal pathway that initiates fate determination and patterning in a gradient-dependent manner, how positional and temporal information is recognised by responding cells remains unknown. Here we have evaluated the role of cell membrane proteins thought to act as obligatory co-receptors for Hh ligand during Ptch-mediated signalling using PGC migration as a model. While our findings are consistent with the role of Ptch1 and Ptch2 in the sequestration of Smo, the reception of Hh ligand and induction of the downstream signalling pathway are mediated distinctly by these two receptors, owing to their specific partnerships with Boc and Gas1. We propose that temporal expression of these receptors determines

**Fig. 6 Gas1 mediates Hh ligand reception by Ptch2. a** Representative images of at least two independent immunoblotting experiments in which NIH3T3/Cas9 cells with *Ptch1* KO were transfected with Myc-tagged *Ptch2* expression construct and stimulated with Shh protein in serum-free medium for 40 min. Interactions of Ptch2 with endogenous Smo, Gas1 and Boc were analysed by immunoprecipitation (IP) and Western blotting (WB). IP was performed using specific antibodies as indicated and analysed by WB with respective antibodies as shown. The images in the bottom panel are from *Ptch1* KO cells transfected with empty vector, which did not show any co-precipitation of endogenous Gas1 and Ptch2 regardless of Shh treatment, confirming the specificity of these antibodies (bottom panel). **b** A schematic diagram illustrating the protein interactions before and after Shh treatment. **c** NIH3T3/Cas9 cells with *Gas1* KO were transfected with Myc-tagged *Ptch2* and stimulated with Shh protein in serum-free medium for 40 min. IP was performed with anti-Myc antibody. Representative blots from two independent experiments are shown. **d** A schematic diagram illustrating the protein interactions in *Gas1* KO cells. Source data are provided as a Source Data file.

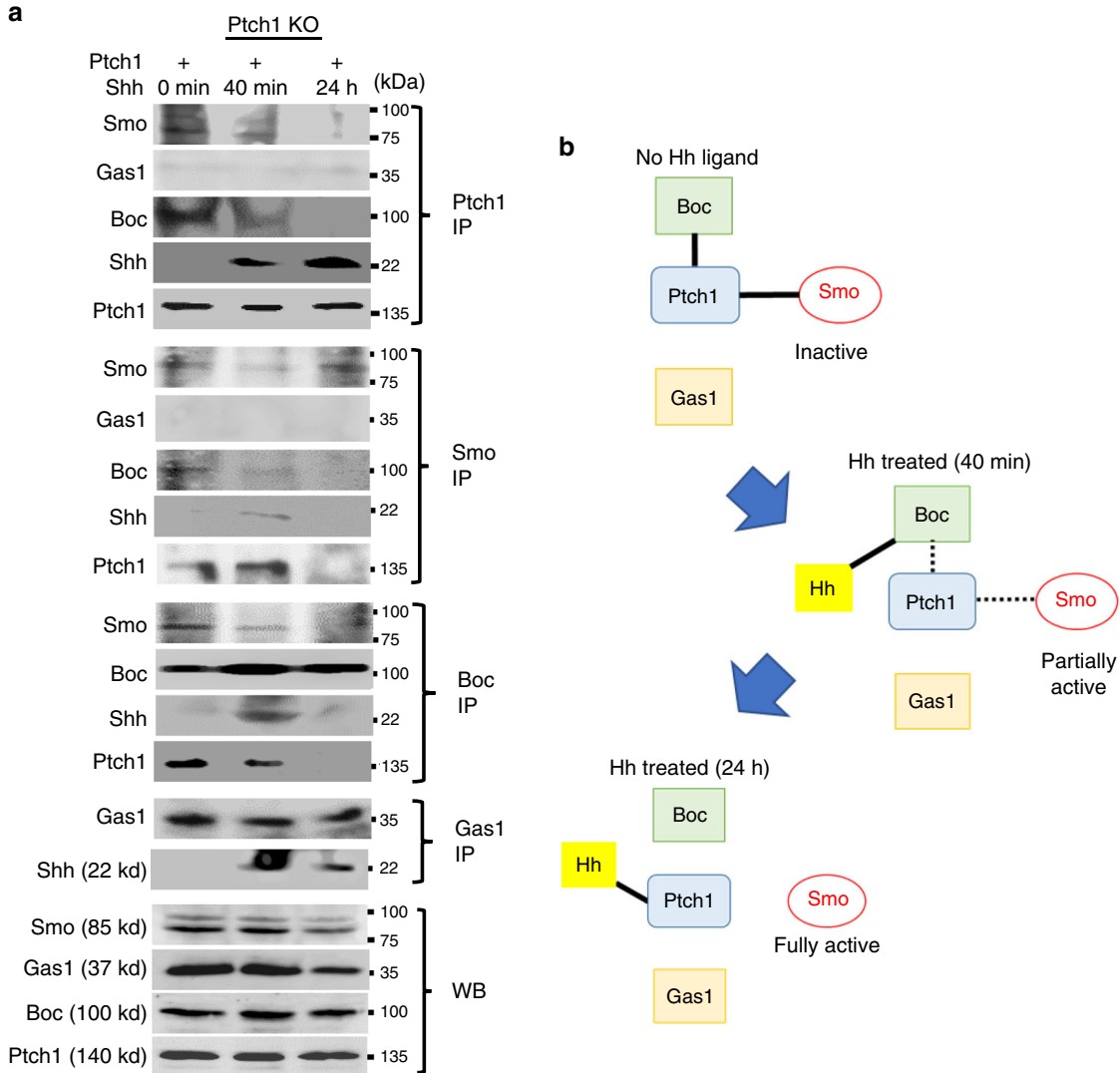

**Fig. 7 Boc mediates Hh ligand reception by Ptch1. a** Representative images of at least two independent immunoblotting experiments in which NIH3T3/Cas9 cells with *Ptch1* KO were transfected with mouse *Ptch1* expression construct and stimulated with Shh for 40 min and 24 h. Interactions of Ptch1 with endogenous Smo, Boc and Gas1 were analysed by IP and WB. IP was performed using specific antibodies as indicated, with WB using respective antibodies as shown. **b** A schematic diagram illustrating the protein interactions before and after Shh treatment. Source data are provided as a Source Data file.

the signalling kinetics in responding cells exposed to a fixed concentration of ligand. It is assumed that the time duration a certain concentration of Shh is available limits the activity and the nature of signalling[53]. In addition to the changing gradient of Shh in vivo, we now reveal that the dynamic flux of ligand presented to the Ptch receptors themselves may be modulated in a context-dependent manner. The changes in the nature of Hh signal may also depend on the availability of the Smo proteins that can be translocated to the cilium upon signalling. Our data suggest that Gas1/Ptch2 may preferentially interact with Smo proteins compartmentalised in the extra-ciliary domain, which are fast-acting to induce motility within a short time scale, such as minutes rather than days after stimulation, as required for the Gli-dependent transcription to occur.

The binding of Shh with Gas1 and Boc co-receptors is likely to create conformational changes in Ptch2 and Ptch1, differentially

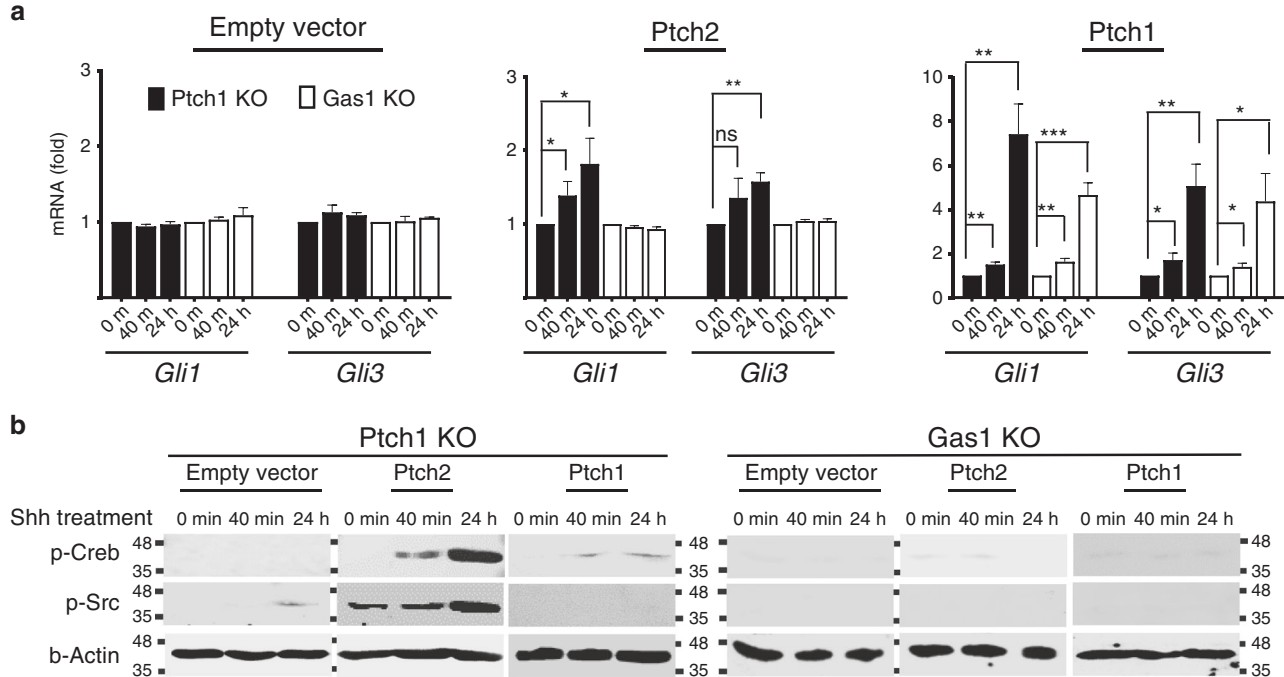

**Fig. 8 Ptch2/Gas1 and Ptch1/Boc induce differential Hh downstream signalling responses. a** NIH3T3/Cas9 cells with *Ptch1* KO or *Gas1* KO were transfected with empty vector, *Ptch2* or *Ptch1* expression construct and stimulated with Shh for 40 min and 24 h. Profile of *Gli1* and *Gli3* transcripts at the respective time points was analysed by quantitative RT-PCR normalised to *Gapdh*. Data represent the mean fold induction compared with the untreated control. Error bars represent SD of three independent experiments. Statistical analysis by unpaired *t* test (*$P < 0.05$, **$P < 0.01$, ***$P < 0.001$). Dot plots showing individual data points are included in the Source Data file. **b** NIH3T3/Cas9 cells with *Ptch1* KO or *Gas1* KO were transfected with empty vector, *Ptch2* or *Ptch1* expression construct and stimulated with Shh as indicated. The signalling responses at the respective time points were analysed by western blotting of total cell lysates using specific antibodies as shown, with b-Actin loading control. Representative blots from three independent experiments are shown. Source data are provided as a Source Data file.

affecting the dissociation of Smo. Ptch1 and Ptch2 share high sequence identity with the luminal and transmembrane domains functionally equivalent in a domain-swapping experiment[15]. However, the cytoplasmic C-terminal tail of Ptch1 required for the repression of Smo, could not be replaced by the analogous C-terminal region of Ptch2[15]. Perhaps the C-terminal tail of Ptch2 may prevent cilia localisation and allow preferential binding with Gas1. Further biochemical, structural and imaging analyses are required to resolve these questions, especially considering the limitation of the current study using overexpressed Ptch and the potential binding of other membrane-associated proteins which we did not investigate. Studies using overexpression constructs provided some initial clues regarding the molecular interactions of these proteins and suggested that Boc may constitutively interact with Ptch1 via its FNIIIa/b domain, but independently interacts with Shh through the FNIIIc domains[33]. The structural domains of Gas1 involved in Ptch2 interaction are unknown, but the core 110 amino-acid tandem repeats were shown to be important in Gas1 interaction with GFRa[54]. Alternatively, Gas1 may simply influence the physical recruitment of partners to the restricted regions of membrane rafts outside the cilium through its GPI-anchor.

We speculate that the rapid release of extra-ciliary Smo proximal to Ptch2/Gas1 may involve different sets of regulatory proteins leading to global cell signalling responses. Although Smo was shown to couple with Gαi, regulating AKT-dependent Gli stabilisation within primary cilia[50], RhoA activation downstream of Smo was independent of AKT or primary cilia, suggesting the existence of ciliary and extra-ciliary pools of Smo exerting separate downstream signalling signatures[49]. Our data support this notion and further demonstrate that extra-ciliary Smo may elicit

parallel signal pathways involving the cAMP-Creb pathway. Although the cAMP-PKA pathway has been majorly explored in Hh signalling, other pathways induced by cAMP can converge to Creb in a PKA-independent manner. Notably, cAMP-induced release of $Ca^{2+}$ leading to the increased phosphorylation of Creb concomitant with downregulation of Gli1 activity was observed during spinal cord development[29]. It is tempting to speculate that such a balance of the Ptch2-mediated p-Creb-dependent parallel pathway may modulate the Ptch1-mediated Gli-driven programs.

We propose that PGC migration depends on local Hh ligand secreted in the proximity of PGCs, which is received by the co-receptor Gas1 and presented to Ptch2. Our data indicate that Hh is not a direct attractant for germ cells but may play important roles in intrinsic motility. The naturally unciliated PGCs are insensitive to the transcription-driven differentiation signal but exhibit migratory responses to Shh. Counter-intuitively, Ptch2 deficient mice are reported to be fertile[21,23]. Although careful assessment data on the potential sub-fertility of these mice are currently unavailable, we speculate that Ptch2 depletion could have been partly compensated by a ligand-independent activation of Smo or its downstream pathway. For instance, there are other signal pathways mediated by different receptors, cytokines and signal transducers known to be essential for the specification, survival and migration of PGCs[2,3]. It is possible that these conserved fundamental developmental signals converge downstream of Smo without the requirement of Ptch2/Gas1. Moreover, Cdon and Boc were shown to mediate chemotactic migration to Shh even when both Ptch1 and Ptch2 were deleted[55], suggesting a possibility that Gas1 induces PGC motility in partnership with other cell surface receptors. It is also worth noting that cells lacking both Ptch1 and Ptch2 retain the ability to migrate toward

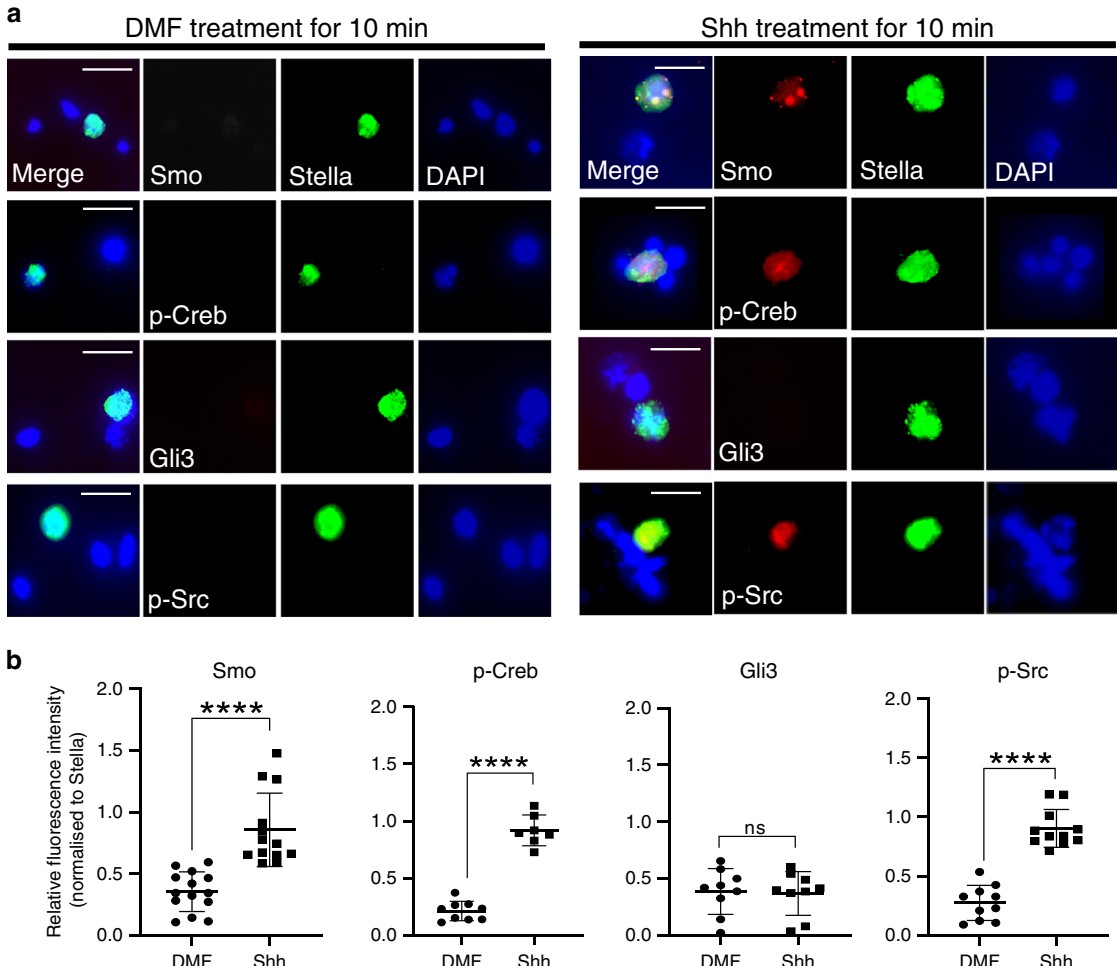

**Fig. 9 PGCs show rapid non-canonical Hh signalling responses. a** Immunofluorescence of Smo, p-Creb, Gli3 and p-Src, co-stained with germ cell marker Stella in GR primary culture cells with or without Shh treatment for 10 min. Representative images from at least three biologically independent experiments are shown. Scale bar, 10 μm. **b** The relative fluorescence intensity values of Smo, p-Creb, Gli3 and p-Src, normalised to the signal intensity of Stella observed in PGCs were compared with or without Shh treatment. Unpaired *t*-test with Welch's correction indicates a significant increase in Smo (DMF ($n = 14$), Shh ($n = 13$), ****$P < 0.0001$), p-Creb (DMF ($n = 9$), Shh ($n = 7$), ****$P < 0.0001$) and p-Src (DMF ($n = 10$), Shh ($n = 11$), ****$P < 0.0001$) but no significant change in Gli3 (DMF ($n = 9$), Shh ($n = 9$), $P = 0.8542$). Error bars represent SD. Unpaired *t*-test with Welch's correction. Source data are provided as a Source Data file.

Shh in a Smo-dependent manner in a Boyden chamber assays[24,56]. It remains to be determined what regulates this PGC-specific expression of Gas1 and other receptors and if Gas1 also influences Hh ligand-specificity, considering the specific expression of Dhh and Ptch2 during genitourinary tract development[16,57]. Our study provides mechanistic insight for PGC migration in mouse and perhaps other species.

## Methods

**Breeding of transgenic mice.** C57BL/6 mice were purchased from Charles River (Harlow, UK). *Stella*^GFP BAC mice[1] were gifted by Professor Azim Surani (Gurdon Institute) and maintained in a C57BL/6 background. *Gas1* mutant mice were originally generated by Dr Chen-Ming Fan (Carnegie Institute of Washington), maintained in a CD1/C57BL6 mixed background and genotyped as previously described[46] using allele specific primers 5′-TACTGCGGCAAGCTTTT CAACGG-3′ and 5′-AGCGCGCTGCTCGTCGTCATATTC-3′ (WT allele); 5′-ACTACGC GTACTGTGA GCCAGAG-3′ and 5′-AGTGACCAGCGAATACCTGTTCC-3′ (*Gas1* mutant allele). For fertility assessment, the number of total litters and litter size of *Gas1*^−/− mice mated with and compared with paired controls were recorded for a period of nine months. All animal experiments were conducted under institutional guidelines under the Animals (Scientific Procedures) Act 1986 in the Biological Research Facilities at St. George's, University of London (PPL 70/8512) and King's College London (PPL7007441).

**Embryo organ culture and live imaging.** Embryo slice culture and filming was conducted as previously reported[1,58] with some modifications. Briefly, the GR of *Stella*^GFP mouse embryos were dissected into the slice culture medium (Hepes-buffered DMEM/F-12 with 0.04% lipid-free BSA and 100 U/ml penicillin/strep-tomycin). The transverse sections were then placed onto millicell CM organ culture inserts (Millipore) pre-coated with mouse collagen IV (50 μg/ml in 0.05 M HCl) (Beckton Dickson) in a 13 mm glass-bottom plate (MatTek, Massachusetts) filled with pre-warmed slice culture medium containing 40 μM of purmorphamine (Cayman Chemical Company), cyclopamine (Stratech Scientific), tomatidine (Sigma-Aldrich) or dimethyl formamide (Sigma-Aldrich), which were optimised for their efficacy with minimal toxicity (Supplementary Fig. 2). The plate was then mounted onto Zeiss LSM510 confocal time-lapse microscope stage in a humidified $CO_2$ chamber at 37.0 ± 0.5 °C. Live imaging was performed at 15 min (0.25 h) interval for a minimum of 10–15 h. Z-stack images of 11–13 μm sections spanning 185–230 μm total thickness were extracted per time point and created into a movie using ImageJ. Velocity was calculated using the formula $V = [sqrt (dx^2 + dy^2)](p)/0.25h$, where dx is the change in the *x*-axis, dy is the change in the *y*-axis and *p* is the pixel size in μm. Typically, for a 10 hour movie, 40 velocity measurements were generated per cell, which was averaged to obtain an overall velocity for that cell. Tracking was performed using ImageJ plugin Chemotaxis and Migration tool (ibidi GmbH, Martinsried, Germany) on the GFP-positive PGCs that remained in focus and viable for the entire duration of the movie.

**Cell cultures.** NIH3T3 cells (American Type Culture Collection, Manassas, VA) were cultured in DMEM containing 2 mM L-glutamine and 100 μg/ml penicillin/

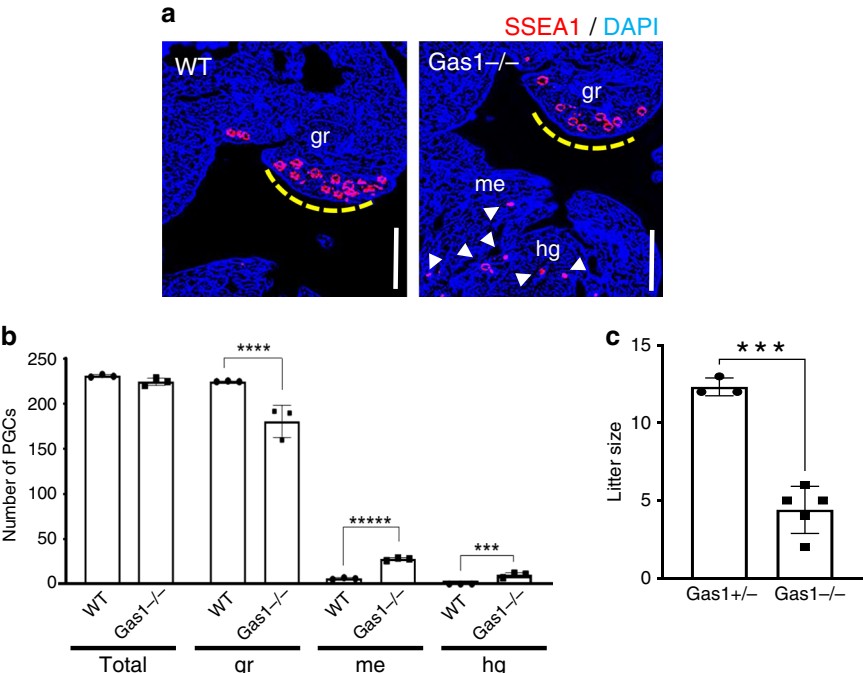

**Fig. 10 Gas1-deficient mice show disrupted PGC migration and sub-fertility. a** A representative image of immunofluorescence staining of E10.5 WT and *Gas1* KO embryos (*n* = 3) using anti-SSEA1 antibody demonstrates the presence of ectopic PGCs (arrows) in the mesentery (me) and hindgut (hg) in *Gas1* KO. Dotted line indicates the genital ridges (gr) where the PGCs should have migrated at this stage of development. Scale bar, 500 µm. **b** Total number of PGCs counted from every other sections of the entire embryo at E10.5 was combined per embryo from each genotype. The numbers of PGCs populated in the genital ridge, mesentery and hindgut are compared between WT and *Gas1* KO. Error bars represent SD from three independent embryos after unpaired *t*-test. ****$P$ = 0.000521 (gr), ****$P$ = 0.000033 (me), ****$P$ = 0.002814 (hg). **c** Litter size is shown as the number of pups per litter for each female caged with a WT male for 9 months. Unpaired *t*-test indicates a significant difference (***$P$ = 0.0002) in the litter size by *Gas1*−/− mice (*n* = 5) compared with *Gas1*+/− (*n* = 3), which were phenotypically normal. Error bars represent SD. Source data are provided as a Source Data file.

streptomycin (Sigma-Aldrich), supplemented with 10% newborn calf serum (NCS). For signalling analyses, NIH3T3 plated at $2 \times 10^6$ cells per 10 cm dish were incubated in 0.5% NCS/DMEM for 24 h and treated with 200 ng/mL recombinant Shh N-terminal peptide[59] (1314-SH, R&D Systems) for 40 min or 24 h. To generate the primary culture of GR cells, the dissected GR tissues were digested in 0.25% trypsin, passed through a 0.4 µm cell strainer and suspended in DMEM/L-15 medium supplemented with 20% knockout serum replacement (Invitrogen), 2 mM L-glutamine, 0.1 mM non-essential amino acids and 0.1 mM 2-mercaptoethanol (Sigma-Aldrich), before being plated onto 0.1% gelatin-coated plates. Cells were incubated in 0.5% serum-containing media before various treatments diluted in serum-free medium.

**Chemotactic migration assay**. Coverslips (8 mm) cut into halves using a diamond blade were coated with 0.001% poly-L-Lysine. For the source of Shh secretion, 293 cells transfected with pCS2 empty vector or encoding full-length human Shh cDNA (a gift from Adrian Salic) were plated without the feeder. For PGCs, the GR cells were plated onto NIH3T3/Cas9 feeder with the targeted knockout of *Wdr11*, which was pre-treated with Mitomycin-C (5 µg/ml) for 2 hours. Once the cells settled down, the two halves of coverslips were carefully rejoined in a 24 well plate containing PGC primary culture medium. After 24 hours incubation, the cells were fixed with 4% paraformaldehyde and stained with anti-SSEA and DAPI. The number of cells present on each side of the coverslips was counted to obtain a percentage of migrated PGCs toward the empty vector or Shh-transfected 293 cells. The distance of PGC migration was measured from the midline.

**Random motility assay**. Dissected GR tissues were prepared to generate single cell suspensions of GR primary culture and plated onto the NIH3T3 feeder layer pre-treated with Mitomycin-C (5 µg/ml) and incubated in 0.5% serum media before treatment with Shh-N, purmorphamine, cyclopamine, tomatidine, DMF (40 µM) and visomodegib (10 µM), all diluted in serum-free medium. Non-directional motility of PGCs was measured by live imaging of GFP-positive cells captured every 15 min for 16 h using Nikon A1R laser scanning confocal microscope in a humidified $CO_2$ chamber at 37.0 ± 0.5 °C. Time-lapse image sequences were analysed using the Chemotaxis and Migration Tool 2.0 plug-in software (Ibidi GmbH).

**Flow cytometry and RNA analyses**. Cells from dissected GR tissues of E10.5 *Stella*^GFP embryos were resuspended in the sorting buffer (1 mM EDTA, 25 mM

HEPES at pH7.0, 1% FBS) and GFP⁺ and GFP⁻ cell population was separated using a MoFlo XDP high-speed cell sorter (Beckman Coulter). FACS data were analysed using FlowJo software. Positive cells had an intensity of >2 on log scale of GFP detection using 488 nm laser for excitation and a filter (529/28 nm) for detection. Cells were selected using FSC (height) versus SSC (height) and a gate was prepared to eliminate debris or abnormally shaped cells. At least 1700 GFP⁺ cells representing ~0.1% of the total sorted cells collected from seven embryos were analysed per experiment. Total RNA from the sorted cells was extracted using PicoPure RNA Isolation Kit (Arcturus) and converted to cDNA using SuperScript IV reverse transcriptase (Invitrogen). Total RNA from mouse tissues was extracted using RNAeasy kit (QIAGEN) and converted to cDNA using NanoScript 2 Reverse Transcription kit (Primer Design). The cDNA obtained was analysed by either GoTaq G2 Hot Start PCR mix (Promega) or Maxima SYBR Green qPCR Master Mix (Thermo Scientific) on a Light-Cycler 2.0 (Roche) using the $2^{-\Delta\Delta CT}$ normalised to *18s rRNA* or *Gapdh*. See Supplementary Table 1 for the primers used.

**CRISPR/Cas9 cell generation and plasmid constructs**. NIH3T3 cells with targeted KO in *Wdr11, Gas1* and *Ptch1* were generated using CRISPR/Cas9 approach. Briefly, sgRNAs designed using the CRISPR Design Tool (http://crispr.mit.edu) were cloned into pSpCas9(BB)-2A-Puro (Addgene #48139) and transfected using Polyfect (Promega). After puromycin (Cambridge Bioscience) selection in 96-well plate, single-cell clones were analysed by Sanger sequencing and western blot to confirm the KO. See Supplementary Table 2 for the sgRNA and primers used. For overexpression in NIH3T3 cells, the vector pcDNA3 (Addgene), pcDNA3-mPtch1 and pCMV6-mPtch2-MycDDK (Origene) were transiently transfected using Polyfect.

**Western blot and co-immunoprecipitation**. Total protein was extracted in a lysis buffer (50 mM HEPES, 150 mM NaCl, 10% glycerol, 1% Nonidet P-40, and 1 mM EDTA) containing protease/phosphatase inhibitors (Sigma-Aldrich), separated by SDS-PAGE, and transferred onto Hybond-ECL membrane (Amersham) which was probed with primary antibodies diluted in blocking buffer (5% skim milk in TBS with 0.05% Tween 20 (TBST)). After washing in TBST, the secondary antibodies conjugated with horseradish peroxidase was added before analyses by enhanced chemiluminescence (GE Healthcare). Co-IP was performed using pre-cleared lysate (1–2 mg protein) and the immune complexes captured on protein A/G-Agarose beads (Santa Cruz Biotechnology) were analysed by Western blot. Uncropped blots are included in the source data file.

**Immunofluorescence and immunohistochemistry**. For tissue staining, embryos at E9.5–10.5 collected after timed-mating were fixed in 4% paraformaldehyde (PFA) before paraffin embedding. Tissue sections were cut at 5–7 μm thickness using an electric microtome (Leica RM2255), deparaffinised with Histoclear (National Diagnostics), and rehydrated in PBS. Following antigen retrieval in Sodium citrate buffer (10 mM Sodium citrate, 0.05% Tween20, pH6.0), sections were blocked with 2% goat serum, 0.2% Triton X-100 in 1xPBS for 1 h at room temperature and then incubated overnight at 4 °C with primary antibodies diluted in antibody dilution buffer (2% goat serum, 0.2% Tween in 1xPBS). After washing in 1xTBST, fluorescence-labelled secondary antibodies were added with counter-staining of DAPI. For immunofluorescence analyses of cells, cultured cells on glass coverslips were fixed with 4% PFA, permeabilized with 0.2% Triton X-100 in PBS, and incubated in blocking buffer (2% heat-inactivated goat serum, 0.2% Triton X-100 in PBS) before probing with primary antibodies diluted in blocking buffer. After washing in 1xTBST, appropriate secondary antibodies were added along with DAPI, before mounting in Mowiol. Secondary antibodies (all from Molecular Probes) were conjugated with Alexa fluor 555 (goat anti-rabbit IgG), Alexa fluor 568 (goat anti-mouse IgG), Alexa fluor 488 (goat anti-rabbit and donkey anti-goat IgG) and used at 1:5000 dilution. For negative controls, non-specific IgG (Sigma-Aldrich) was used instead of primary antibodies. Fluorescence microscopy was performed using Zeiss Axioplan 2 Upright and analysed by using Fiji ImageJ software (National Institutes of Health). For quantification, local background subtraction was performed before analysis.

**Antibodies**. Antibodies were generated against Myc (M4439, mouse monoclonal 1:200, Sigma-Aldrich), Ptch1 (sc-293416, mouse monoclonal 1:200, Santa Cruz), Ptch2 (PA1-46223, rabbit polyclonal 1:200, Invitrogen), Gas1 (AF2636, goat polyclonal 1:500, R&D; PA5-48298, rabbit polyclonal 1:500), Boc (AF2385, goat polyclonal 1:500, R&D; MAB20361, mouse monoclonal 1:200, R&D), Cdon (GTX105422, rabbit polyclonal 1:500, GeneTex), Smo (sc-166685, mouse monoclonal 1:200, Santa Cruz, A3274, rabbit polyclonal 1:200, ABclonal), Shh (sc-365112, mouse monoclonal 1:200, Santa Cruz), Gli3 (AF3690, goat polyclonal 1:200, R&D), WDR11 (ab175256, rabbit polyclonal 1:200, Abcam), SSEA1(MC-480, mouse monoclonal 1:200, DSHB), Stella (ab19878, rabbit polyclonal 1:200, Abcam), phospho-Src, (sc-166860, mouse monoclonal 1:500, Santa Cruz), phospho-Creb (sc-81486, mouse monoclonal 1:500, Santa Cruz), Arl13b (17711-1-AP, rabbit polyclonal 1:500, Proteintech), IFT88 (13967-1-AP, rabbit polyclonal 1:500, Proteintech), CEP164 (sc-515403, mouse monoclonal 1:200, Santa cruz), gamma-tubulin (T6557, mouse monoclonal 1:500, Sigma), b-actin (4967L, rabbit polyclonal 1:500, CST). HRP-conjugated secondary antibodies used were goat anti-mouse IgG (sc-516102, 1:1000, Santa Cruz), goat anti-rabbit IgG (ab6721, 1:1000, Abcam), donkey anti-goat IgG (ab6741, 1:1000, Abcam).

**Statistical analysis**. Data were analysed by the investigator blinded to the experimental condition, using GraphPad Prism 6 (La Jolla, CA, USA). Statistical significance was determined by unpaired two-tailed Student's $t$ test with Welch's correction and one-way analysis of variance (ANOVA) followed by Dunnett's test.

**Reporting summary**. Further information on research design is available in the Nature Research Reporting Summary linked to this article.

## Data availability

The authors declare that all data supporting the findings of this study are available within the article and its supplementary information files or from the corresponding author upon reasonable request.

The source data underlying Figs. 2b–d, 3b–d, 4b, d, 5b, d, 6a, c, 7a, 8a, b, 9b, 10b, c and Supplementary Figs. 2a–d, 6, 7 and 8b are provided as Source Data files.

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

## Acknowledgements

Y.J.K., J.Y.L. and S-H.K. were supported by Medical Research Council (MRC) grant MR/L020378/1 awarded to S-H.K. J.Y.L. and Y.J.K. were also supported by St George's Research Bridging Fund Scheme. M.S. was supported by a European Union Marie Curie Early Stage Fellowship (MEST-CT-2004-504025) and an Academy of Medical Sciences Starter Grant for Clinical Lecturers. M.S. and M.T.C. were also supported by the European Orthodontic Society. We are grateful for Azim Surani and Chen-Ming Fan for the transgenic mouse lines, Adrian Salic for the SHH-expression construct and Christopher Wylie for the embryo slice culture techniques. We thank Isabelle Crevel, Joanna Nolan and Gregory Perry for technical assistance with cell sorting and live imaging microscopy.

## Author contributions

Conceptualisation, funding acquisition and supervision: S-H.K; Original draft writing: S-H.K; Methodology (creation of models and design of methods): Y.J.K., J.Y.L., S-H.K., M.S., M.T.C.; Investigation (performing the experiments): Y.J.K., J.Y.L., M.S.; Manuscript review and editing: Y.J.K., J.Y.L., M.T.C. and S-H.K.

## Competing interests

The authors declare no competing interests.
