## [Peer Review File · Nature Communications]

Reviewers' Comments:

Reviewer #1:

Remarks to the Author:

This paper puts forward the intriguing possibility that dependent on the use of distinct Shh co-receptors (Boc and Gas1) in conjunction with signaling via Ptch1 or Ptch2 Smo activation can result in distinct signaling outcomes. Evidence is provided using a combination of PGC migration experiments, and hypotheses are further tested in cell lines using biochemical approaches. The observation that PGC only express Ptch2 and not Ptch1) is highly intriguing and possibly highly useful in assessing distinct functions of Ptch1/2 and Boc/Gas1. It remains largely unclear how the biochemical findings in 3T3s are informative to the perhaps unique interaction between Shh/Gas1/Ptch2 in PGCs.

The conclusions regarding the lack of directed migrations in the co-culture experiments using 293 cells might be reasonable. However, it is inappropriate to equate the Purmorphamine activity (presumably via the sterol binding pocket of Smo) with that of Shh, as the non-canonical Hh response requires the CRD of Smo. The Purmorphamine experiments are further complicated by the experimental approach that presumably provides Purmorphamine at a given flat concentration, and not in a gradient, as would be formed in a Boyden chamber or equivalent device. Furthermore, the use of 40 μ M cyclopamine is very high and cyclopamine becomes toxic at low μ M concentrations, in particular in primary cells. The migration speed experiments would be a lot more convincing if Shh was used as the ligand and the Smo requirement was addressed using Vismodegib, or if so needed, with cyclopamine at a high nanomolar concentrations. Similar arguments hold for the results shown in 4A. They should be repeated using ShhN, and with ShhN plus Vismodegib/cyclopamine. See also comments about Fig 7C.

Migration/speediness by PGCs derived from Ptch2 null mice must be assessed to address the requirement and in vivo specificity for the proposed interactions.

I highly appreciate the use of Ptch1 and Gas1 knockout 3T3 cells. But this immediately raises the question why no Ptch2 null line was made for these experiments, and to some extent Boc. I quote "We used NIH3T3 cells as a well-established model, which normally express Ptch1, but not Ptch2." This statement is not supported with any experimental data. As Ptch2 is commonly activated in cells with an upregulated Shh response as is predicted in the cells lacking Ptch1. Furthermore, roles of Ptch2 only becomes possible to assess in the absence of Ptch1. Ideally the Ptch1/Ptch2 transfection experiments should be done in Ptch1/2 double null cells. Furthermore, the Gli3/GliR blots look puzzling, and only partially quantified and invisible in lanes for no apparent reason. If conclusions need to be derived from those, at least some statistics of multiple experiments should be shown.

For the IP some of the controls are sparse, in particular for the no-bait (ie Ptch2 and Gas). The Boc IPs for Ptch2 need to be shown as well. The WB for ShhN should be shown. If the authors want to propose a Ptch1/2-mediated "sequestration" of Smo, then at a minimum the inverse IPs need to be shown. Co-IPs of large multispan proteins is not only fraud with artifacts but must be interpreted in the light that the functional interactions between Ptch1 and Smo are catalytic and not stoichiometric and can even be non-cell autonomous.

Fig 7C. I induction of p-CREB and p-Src is very striking, and in many ways be best biological readout provided. Does Ptch2/Gas1 expression (after transfection) in soma cells (or another appropriate cell) allow Shh-induced p-Creb and p-Src induction? Why does Smo become visible 10 min after Shh/Purmorphamine treatment? As Stella is a guide for PGS, it should be pretty straightforward to some numbers on the efficiency of this response. It would also be straightforward to perform these experiments with Purmorphamine to buttress the findings in Fig 3 and 4A.

Gene names should be italicized.

It should be made clear what bands on gel are Shh or ShhN

Reviewer #2:

Remarks to the Author:

This is an interesting study which considers the role of Hedgehog signalling in potentially regulating primordial germ cell (PGCs) migration, at the point where those cells exit the hindgut and colonise the genital ridges. In migrating PGCs, and PGCs that have just colonized the genital ridges, heter-complexes of Ptch2/Gas1 seem important whilst surrounding somatic cells express Ptch1/Boc hetero-complexes. Using NIH3T3 cells, Hh signalling through the former receptor complex was associated with rapid p-Src and p-Creb induction, a previously unreported Hh signalling response. In the same NIH3T3 system, signalling through the Ptch1/Boc receptor complex lead to the canonical Hh response of upregulated Gli activity. The study also reveals that PGCs are naturally unciliated, and so the normal translocation of released Smo to the cilia is not possible. The results will add significantly to our understanding of the breadth of possible Hh signalling responses and their regulation.

Major comments

What about the co-receptor CDON? You don't find it expressed in Figure 1B, but other results would suggest it is actually highly expressed at least at E11.5 (Jameson et al, 2012 PLoS Genet. - microarray data). Do you know your primers work?

I am not sure why you say 'PGCs exclusively express Ptch2 and Gas1 while ciliated somatic cells immediately surrounding PGCs express Ptch1 and Boc' (line 129)?? I think Figure 1D and Figure 5C suggest that Boc is also expressed by PGCs?

Figure 5A – you isolate PGCs and somatic cells from GR by FACS. Why are these cells isolated from 'GR cultures' (not from freshly-dissected GRs?). More importantly, I think that you could and should address cell type specificity of expression of Gas1, Boc (and Cdon) in these samples.

I don't see any reason why the analysis of Gas1-null mice should be relegated to the Supplementary information.

The Ptch2 null mouse model is reportedly fertile (Lee et al, 2006 – your ref 34). This one seems a complete KO (while the one in ref 43 is a hypomorph). What does this mean in terms of Hh regulation of germ cell migration, particularly when considered in combination with the Gas1 null phenotype? Can you comment on this please in the Discussion.

The topic is complex but, nonetheless, the Introduction seems extremely long.

Minor points

Very weird to have a list of reference in the first paragraph as 23,31,49,52 (presumably these should be 1,2,3,4 and the Reference list amended accordingly).

Line 63 – the word 'demonstrated' should probably be replaced by 'claimed' since the reports present conflicting views

In Figure 4c, the numbers 1 and 2 seem to be missing from the first panel

Please note the timepoint at which PGCs were isolated from Stella-GFP embryos (line 218), Figure

5 legend, Supp Fig 2 legend. This appears only in the Methods (E10.5).

Saying the 'anlage of E9.5-11.5 embryos' doesn't really make sense (line 147). You should say 'gonadal anlage'?

The videos are not labelled as S1, S2, S3, S4 – I assumed they are in numerical order (4033805, 6, 7, 8)

Reviewer #3:

Remarks to the Author:

Review report for manuscript NCOMMS-19-28270 entitled "Ptch2/Gas1 and Ptch1/Boc differentially regulate Hedgehog signalling in primordial germ cell migration"

In this manuscript, the authors report differential pathways of Hedgehog (HH) signaling mediated by receptor complexes, Ptch2/Gas1 and Ptch1/Boc. At the primordial germ cell (PGC)-migration stages in mouse development (E10.5-E11.5), the authors report that Ptch2 and Gas1 are expressed only in PGCs, marked by Stella-EGFP or SSEA1, while Ptch1 and Boc are expressed in somatic cells. The HH signaling seems to enhance migratory activity of PGCs in embryonic slice cultures *ex vivo*, without affecting their directionality. Interestingly, Boc and Gas1 seems to be differentially required in the HH signaling pathways mediated by Ptch1 and Ptch2, respectively. The authors also report that PGCs do not have mature primary cilia, but PGCs activate Smo and Gli3, known downstream factors of the HH signaling associated with the cilia, in a different manner from the canonical pathway. Consistently, the authors describe that Gas1 mutant embryos show migratory defect of PGCs. Together, the concept of the alternative/non-canonical HH signaling specific to PGCs may be interesting. However, I have many significant concerns with on the data quality and data interpretations in this manuscript.

(1) Immunofluorescence analyses.

The quality of immunofluorescence (IF) is not very high. Even in the representative images, the fluorescence signals are often saturated perhaps due to too high signal gains (e.g., Fig 1D, SSEA1; Fig 5B DAPI). The IF images also often show too high background (e.g., Fig 5C DAPI). Such signal saturation and/or high background make it difficult to evaluate the IF data.

Moreover, many IF panels show only one or two PGCs or somatic cells in the field of view with poor information of surrounding cells (e.g., Fig 4A and others). Because development proceeds with cellular interactions, such poor histological information also makes it difficult to evaluate the IF data. In addition, this study often relies solely on visual inspection of representative images for the analyses of the IF data, and lacks statistical analyses.

(2) Specific expression of Ptch2 and Gas1 in PGCs

The authors claim that Ptch2 and Gas1 are specifically expressed in PGCs but not in somatic cells (Fig 1D, 5A, 5B), and that they function in the PGC migration in a cell-autonomous manner. However, in Fig 1C, Gas1 is clearly expressed in somatic cells in hindgut, dorsal mesentery, and intermediate mesoderm at E9.5 and/or E10.5. All of these tissues are along the route of the PGC migration, and thus, Gas1 may function indirectly in the PGC migration.

In addition, in Fig 5A and 5B, two different methods are used to evaluate the expression of Ptch1/Ptch2 and Gas1/Boc; Ptch1 and Ptch2 are analyzed using real-time PCR (Fig 5A) while Gas1 and Boc are analyzed using IF (Fig 5B). This is strange to me. Why do not the authors use the same method consistently for these factors? (i.e., IF for both or real-time PCR for both).

Moreover, for the real-time PCR analysis, according to Materials and Methods, $2^{\Delta\Delta CT}$ values are plotted and tested in Fig 5A. However, because the raw data of real-time PCR are CT

values, statistical tests should be also performed for CT values or their linear derivatives (e.g., delta-delta CT values).

(3) Migration assay of PGCs in the slice culture

The authors assay migratory activity of PGCs using embryonic slice cultures exposed to agonist and antagonist of HH (Fig 2 and 3). By tracing the stella-EGFP-positive PGCs, the authors conclude that the HH signaling enhances the PGC migration in a non-directional manner. I failed to find the information how the authors chose the concentrations of the agonist/antagonist (40uM).

Moreover, the embryonic slices cultured with the agonist/antagonist rapidly proliferate or grow with very different kinetics accompanying macroscopic morphological changes as shown in the supplementary videos. Thus, the agonist/antagonist clearly act on the surrounding somatic cells, and the change of the migration speeds of PGCs may be an indirect and/or apparent effect.

In addition, if the HH signaling enhances the PGC migration, it would be expected that PGCs may form more lamellipodia and/or filopodia. Thus, cellular morphology of PGCs under influence of the HH signaling should be investigated.

(4) Fig 4A: Cytoplasm localization of Smo and Gli3

The authors report that the protein levels of Smo and Gli3 increase in cytoplasm when PGCs are treated with a HH agonist (Page 9, Line 207) (Fig 4A). However, Fig 4A clearly shows that signals of Smo and Gli3 are overlapped with the DAPI signal, contradicting the authors' claim.

(5) Fig 4B: Quantification of IF intensities

The IF intensities of Smo and Gli3 are quantified and compared between PGCs treated with purmorphamine (PUR) and DFM (DMF (dimethyl formamide)?) (Fig 4B). However, quantitative measurement of fluorescence intensity of IF is generally difficult and needs careful interpretation. For example, in order to control the variation of staining, the two cell types of which IF intensities are compared (i.e., +DFM and +PUR cells) should be intermingled on the same slide glasses and stained at the same time.

(6) Fig 4C: un-ciliated PGCs

The authors report that PGCs are un-ciliated based on the lack of the dots of Arl13b and Ac-T, markers for primary cilia (Fig 4C). However, Fig 4C also shows that there are somatic cells negative for the dots of the cilia markers. Thus, it is required to show statistics of PGCs and somatic cells that are positive/negative for these markers.

(7) Fig 5C: "E13.5 PGCs".

The authors analyze the p-Src level in "E13.5 PGCs" to evaluate the Sonic Hedgehog (Shh) signaling mediated by the Ptch2/Gas1 complex in non-migratory germ cells (Fig 5C). However, I failed to identify the sex of these germ cells: At E13.5, mouse gonads clearly show a morphological difference between male and female, and germ cells also undergo clear sex differentiation (mitotic arrest in male, and meiotic entry in female).

Moreover, because E13.5 germ cells are at quite distinct cell-cycle phases (mitotic arrest or meiosis) from the E10.5 PGCs (mitotically active), I think that it may not be appropriate to argue that Ptch2/Gas1 is functionally important for migration because p-Src is positive in E10.5 PGCs and is negative in E13.5 germ cells.

(8) Fig 6 and 7: Over-expression of Ptch1 and Ptch2 in the Ptch1-KO NIH3T3 cells

Using Ptch1/Ptch2-over expressing cells, the authors perform Co-IP analyses for the Ptch1/Ptch2 proteins to assay the binding activities of Gas1, Boc, and Shh to these receptors (Fig 6A, 6C and 7A). The authors also assay the levels of downstream factors, pCreb, pSrc, and the Gli3 cleavage (Fig 6E). However, I failed to find the methods and primary results of the over-expression. Which promoters are used, and how? Are Ptch1/Ptch2 expressed transiently or stably? What are the

expression levels of the over-expressed proteins? Are the over-expression levels comparable to endogenous/physiological levels in NIH3T3 cells and/or PGCs? Are the over-expression levels of Ptch1 and Ptch2 similar to each other sufficiently for the comparison of the signal transduction activities? All of the above are critical for the appropriate evaluation of the experiments described in this manuscript.

(9) Fig 6E: Gli3 FL/R ratio.

The authors report that Ptch1 and Ptch2 show different kinetics of stabilization of Gli3 protein in response to Shh signaling based on the ratio of full-length and cleaved Gli3 (Gli3 FL and Gli3 R, respectively) on the Western blot analyses (Fig 6E). However, the total protein level of Gli3 seems to change significantly upon the Shh treatment; e.g., in the Ptch2 over-expressing cells, the Gli3 level decreases transiently at 40 min after the Shh treatment and then drastically increases at 24 hours. With such a change of the protein level, it may not be appropriate to simply interpret the Gli3 FL/R ratio as an indicator of the Gli3 stabilization level. Moreover, the quantification of the Gli3 FL/R ratio should be repeated for several times with a statistical test.

In addition, the logic of the interpretation of the Gli3 FL/R ratio is not clear to me: this ratio changes from 0.3 to 0.6 (twice) in the Ptch2 over-expressing cells at 40 min, while it changes from 0.2 to 0.7 (3.5 times) in the Ptch1 over-expressing cells at the same time (Fig 6E, left panels). With a hypothesis that this ratio indicates the Gli3 stabilization level (i.e., HH signaling level), the authors claim that Ptch2 transduces the HH signaling rapidly, while Ptch1 transduces it gradually. However, simply based on the Gli3 FL/R ratio above, the following interpretation is also possible; Gli3 is more rapidly stabilized in the Ptch1 over-expressing cells, and this may lead us to speculate that Ptch1 transduces the HH signaling more rapidly, and the signal transduction continues to be reinforced for at least 24 hrs, even after Boc is released from Ptch1 (Fig 7A), while Ptch2 transduces it more slowly and less efficiently thereby reaching the plateaux at a low level at short times.

(10) PGCs in the Gas1-knockout mice

The authors report a migratory delay and existence of ectopic PGCs in the Gas1-mutant embryos (Fig S5A, S5B). However, Gas1 deficiency may affect PGC specification or survival, and thus, total number of PGCs in the mutant embryos should be counted.

We respond to each of the reviewers' specific points as below.
The changes are also highlighted in the main text.

In response to Reviewer #1:

This paper puts forward the intriguing possibility that dependent on the use of distinct Shh co-receptors (Boc and Gas1) in conjunction with signaling via Ptch1 or Ptch2, Smo activation can result in distinct signaling outcomes. Evidence is provided using a combination of PGC migration experiments, and hypotheses are further tested in cell lines using biochemical approaches. The observation that PGC only express Ptch2 and not Ptch1 is highly intriguing and possibly highly useful in assessing distinct functions of Ptch1/2 and Boc/Gas1. It remains largely unclear how the biochemical findings in 3T3s are informative to the perhaps unique interaction between Shh/Gas1/Ptch2 in PGCs.

1) The conclusions regarding the lack of directed migrations in the co-culture experiments using 293 cells might be reasonable. However, it is inappropriate to equate the Purmorphamine activity (presumably via the sterol binding pocket of Smo) with that of Shh, as the non-canonical Hh response requires the CRD of Smo.

=> We now include new motility data from live-imaging of PGCs in the genital ridge (GR) primary cultures which are treated with Shh-N, purmorphamine, cyclopamine, vismodegib and tomatidine (Figure 2D & Supplementary Movie 5-10). The data show that both Shh-N and purmorphamine significantly increased the intrinsic motility of PGCs. As the Reviewer rightly pointed out, Shh-N showed a slightly higher efficiency compared to purmorphamine, indicating their different mechanism of action. Nonetheless, the data clearly support the notion that the intrinsic motility of PGCs is Hh-dependent. In addition, treatment with vismodegib and cyclopamine consistently resulted in an inhibitory effect on PGC motility. We found no visible cytotoxicity during the imaging of these cells compared to the control group. These data unambiguously confirm our original observation, which is reproducible using complementary assays.

2) The purmorphamine experiments are further complicated by the experimental approach that presumably provides purmorphamine at a given flat concentration, and not in a gradient, as would be formed in a Boyden chamber or equivalent device.

=> Since we did not alter the endogenous Hh expression in these embryos, the innate gradient of Hh within this region (as shown in Fig 1C & E) is preserved and remains operative in our slice culture experiments. This is further evidenced by the fact that Hh agonist/antagonist treatment did not alter the directionality of migration. Previous studies in other species have reported that PGCs do not follow a Hh gradient (references 6-8). Combined with our own chemotaxis data (Fig 2 A-C) and new motility analyses *in vitro* (Fig 2D), the embryo slice culture represents a powerful tool to demonstrate the behaviour of PGCs *ex vivo* within their native environment, including the normal extracellular matrix, which is lacking in a Boyden chamber setting.

3) Furthermore, the use of 40 μ M cyclopamine is very high and cyclopamine becomes toxic at low μ M concentrations, in particular in primary cells.

=> Embryo organ cultures are more resistant to drugs as it needs to diffuse through the tissues. At least in two previous publications where E10.5 mouse embryo organ cultures were similarly used, cyclopamine was shown to be effective at 20 μ M or above (Lipinski et al., *Toxicol Sci.* (2008) 104:189–197 & Nagase et al., *J Craniofac Surg.* (2005) 16:80-8). Based on these papers, we assessed the effectiveness of cyclopamine in our slice cultures using a range of concentrations. We did not see any effects up to 20 μ M but started seeing a reduction in PGC motility at 40-60 μ M without notable cytotoxicity. We only noticed some toxicity when 80 μ M was applied, at which even the solvent control showed some slight toxicity. We similarly screened varying concentrations of

purmorphamine and found 40-60 μM effective (Please see Supplementary Figure 2 A-C). To quantitatively assess any general toxicity on the overall survival of PGCs (GFP-positive), we measured the number of hours that the green fluorescence signal could be observed during the live-imaging and found no difference in the cyclopamine-treated group (Fig 3C). After these optimisations, we chose to use 40 μM in our experiments. We do appreciate the importance of the reviewer's point and include further text regarding the toxicity of these drugs in the main text (page 7, line 164).

4) The migration speed experiments would be a lot more convincing if Shh was used as the ligand and the Smo requirement was addressed using vismodegib, or if so needed, with cyclopamine at a high nanomolar concentrations. Similar arguments hold for the results shown in 4A. They should be repeated using ShhN, and with ShhN plus vismodegib/cyclopamine. See also comments about Fig 7C.

=> We have now included PGC motility data using the GR primary cultures which are treated with Shh-N, purmorphamine, cyclopamine, vismodegib and tomatidine (Figure 2D). We tested different concentrations of vismodegib (0, 5, 10 & 20 μM) based on previous publications, and found no general toxicity (Supplementary Figure S2D). We chose to use 10 μM vismodegib as it was the lowest concentration that caused a statistically significant difference in motility.

=> Please note that Shh-N was already used in the original Figure 7C (Figure 9A in the revision). Since the GR cultures exhibit very little basal level expression of p-Creb, p-Src or Gli3 in the absence of Hh stimulation, we believe the current data reliably represents active induction of the Hh signalling pathway.

5) Migration/speediness by PGCs derived from *Ptch2* null mice must be assessed to address the requirement and *in vivo* specificity for the proposed interactions.

=> The suggested *in vivo* experiment requires generation of a new transgenic hybrid mouse line, i.e. Stella-GFP in a *Ptch2*^{-/-} background, which is beyond the time-frame of this revision. However, our current data demonstrate that PGCs in a *Gas1*^{-/-} background still express *Ptch2* (Fig 5C) but show reduced migration *in vivo* (Fig 10A), supporting the notion that co-expression of *Ptch2* and *Gas1* is required for appropriate migration of PGCs. However, we do appreciate the importance of the reviewer's point and include further text regarding the fertility phenotypes of *Ptch2*^{-/-} mice in the Discussion (page 17, line 402-412).

6) I highly appreciate the use of *Ptch1* and *Gas1* knockout 3T3 cells. But this immediately raises the question why no *Ptch2* null line was made for these experiments, and to some extent Boc. I quote "We used NIH3T3 cells as a well-established model, which normally express *Ptch1*, but not *Ptch2*." This statement is not supported with any experimental data.

=> We already confirmed the lack of endogenous *Ptch2* expression in WT NIH3T3 cells by RT-PCR (Supplementary Fig S5) and Western blot (Supplementary Fig S7A) but these data were omitted from the original submission. We now include this important information.

7) *Ptch2* is commonly activated in cells with an upregulated Shh response as is predicted in the cells lacking *Ptch1*. Furthermore, roles of *Ptch2* only becomes possible to assess in the absence of *Ptch1*. Ideally the *Ptch1*/*Ptch2* transfection experiments should be done in *Ptch1/2* double null cells.

=> As mentioned above, our *Ptch1* KO NIH3T3 cells are in fact *Ptch1/Ptch2* double null. To our knowledge, no published data to date has demonstrated that loss of *Ptch1* results in full activation of the Hh pathway independent of ligand stimulation. A modest increase (2-3 fold) of *Ptch1* and *Gli1* at transcription level was reported in *Ptch1*-deficient mouse epidermis (*Journal of Investigative Dermatology* (2017) 137:179e186) and in a basal cell carcinoma-derived model (*Cellular Oncology* (2018) 41:427-437). These studies used developing hair follicle cells or skin cancer cells, which were likely to be "primed" by the specific fate determination or oncogenic environment. Notably, despite their presumably higher baseline, these *Ptch1*-deficient cells still responded to the ligand/agonist

treatment in a dose-dependent manner. In our *Ptch1* KO NIH3T3 cells, we did not see a uniformly high level of constitutive canonical Hh signalling compared to the control, possibly reflecting the nature of NIH3T3 *in vitro*, as they do not secrete a very high level of Hh ligand that could cell-autonomously activate the pathway.

8) Furthermore, the Gli3/GliR blots look puzzling, and only partially quantified and invisible in lanes for no apparent reason. If conclusions need to be derived from those, at least some statistics of multiple experiments should be shown.

=> We now include quantitative RT-PCR results, which assess *Gli1* and *Gli3* transcription profiles, a classical quantitative readout for Hh signalling. The data over the key time-points with statistical analyses demonstrate that *Ptch1* and *Ptch2* receptors show different signalling capacity (Fig 8A). The results are fully discussed in the main text (page 12 - 13).

9) For the IP, some of the controls are sparse, in particular for the no-bait (ie *Ptch2* and *Gas1*).

=> We now include the Co-IP Western blot data from empty vector-transfected *Ptch1* KO cells after *Ptch2* IP and *Gas1* IP (supplementary Fig S7B). Additional reverse Co-IP data are also included in Fig 6A and 7A.

10) The WB for ShhN should be shown.

=> We now include the Western blot data (Fig 6A, the last image). As the very faint band on the gel indicates, NIH3T3 cells do not produce a high level of endogenous Shh. We could not detect the exogenously added recombinant Shh-N protein in Western blots using total cell lysates. However, our Co-IP did show the presence of Shh-N bound to the receptors, likely due to concentration of the protein in the immunocomplex.

11) The Boc IPs for *Ptch2* need to be shown as well.

=> We now also include Boc Co-IP Western blot data, confirming the specific interaction of Boc with *Ptch1*, but not with *Ptch2* (Fig 6A and 7A).

12) If the authors want to propose a *Ptch1/2*-mediated “sequestration” of Smo, then at a minimum the inverse IPs need to be shown.

=> We now include Smo Co-IP Western blot data, confirming the dynamic interactions of Smo with different receptors (Fig 6A and Fig 7A).

13) Co-IPs of large multispan proteins is not only fraught with artifacts but must be interpreted in the light that the functional interactions between *Ptch1* and Smo are catalytic and not stoichiometric and can even be non-cell autonomous.

=> We agree with the Reviewer’s point regarding the application of caution for potential artefacts in this type of assay. Therefore, we intentionally avoided overexpression experiments, such as co-transfecting multiple genes as done in other previous publications. Except for *Ptch1/2*, we analysed only endogenous proteins, thus allowing the natural stoichiometry between binding partners. Our Co-IP data also demonstrate that the binding interaction between *Ptch1* and Smo is transient and time-dependent, potentially indicating a catalytic nature. In addition, we studied the intracellular co-localisation of these proteins by immunofluorescence in primary cultures where the individual PGCs and somatic cells are isolated from each other in single-cell suspensions (Fig’s 2, 4, 7), so it is most likely that we are observing cell autonomous effects. Our complementary approaches thus provide comprehensive evidence for genuine interactions between these proteins.

14) Fig 7C. Induction of p-CREB and p-Src is very striking, and in many ways the best biological readout provided. Does *Ptch2/Gas1* expression (after transfection) in soma cells (or another appropriate cell) allow Shh-induced p-Creb and p-Src induction?

=> The somatic cells surrounding PGCs express endogenous *Ptch1* (Fig 5A & B). Transfecting *Ptch2* into those cells (or any other cell types with endogenous *Ptch1*) will not show clear *Ptch2*-dependent responses, as they are obscured by the effects of *Ptch1*. This is why we used the NIH3T3 cell system after *Ptch1*-deletion to mimic the native context of PGCs as closely as possible for the investigation of *Ptch2*/*Gas1*-specific signalling.

15) Why does Smo become visible 10 min after Shh/Purmorphamine treatment?

=> Smo is not transcriptionally regulated by Hh signalling. Upon de-repression from *Ptch1*, Smo is believed to undergo phosphorylation by GRK2, leading to protein stabilisation and accumulation on the plasma membrane. Our data consistently showed that addition of Hh ligand or purmorphine up-regulates the level of extra-ciliary Smo protein in PGCs. Whether this is through blocking the β -arrestin2-mediated internalisation and degradation of Smo after activation is not clear. Nonetheless, this behaviour is clearly distinguishable from the ciliated somatic cells where Smo translocates to the cilium. Therefore, the sub-cellular location of Smo appears to be important for its activation mechanism. The plasma membrane may provide a favourable environment for Smo to be accessible to non-ciliary intracellular signal transducers, which will need further investigation in the future.

16) As Stella is a guide for PGS, it should be pretty straightforward to some numbers on the efficiency of this response. It would also be straightforward to perform these experiments with Purmorphamine to buttress the findings in Fig 3 and 4A.

=> We now present the efficiency of the responses as a graph with appropriate statistical analyses (Fig 9B).

17) Gene names should be italicized.

=> We have italicised the gene names throughout the revised text.

18) It should be made clear what bands on gel are Shh or ShhN

=> We could not clearly detect the endogenous Shh or Shh-N in normal Western blot using total cell lysate (see statement above). The Shh band visible after Co-IP is Shh-N as the molecular weight indicates in the figure.

In response to Reviewer #2:

This is an interesting study which considers the role of Hedgehog signalling in potentially regulating primordial germ cell (PGCs) migration, at the point where those cells exit the hindgut and colonise the genital ridges. In migrating PGCs, and PGCs that have just colonized the genital ridges, hetero-complexes of *Ptch2*/*Gas1* seem important whilst surrounding somatic cells express *Ptch1*/*Boc* hetero-complexes. Using NIH3T3 cells, Hh signalling through the former receptor complex was associated with rapid p-Src and p-Creb induction, a previously unreported Hh signalling response. In the same NIH3T3 system, signalling through the *Ptch1*/*Boc* receptor complex lead to the canonical Hh response of upregulated Gli activity. The study also reveals that PGCs are naturally unciliated, and so the normal translocation of released Smo to the cilia is not possible. The results will add significantly to our understanding of the breadth of possible Hh signalling responses and their regulation.

Major comments

1) What about the co-receptor CDON? You don't find it expressed in Figure 1B, but other results would suggest it is actually highly expressed at least at E11.5 (Jameson et al, 2012 PLoS Genet. - microarray data). Do you know your primers work?

=> Yes, we did validate our *Cdon* primers in NIH3T3 cells (Supplementary Fig S5B). Our immunofluorescence data further show that *Cdon* protein is not detected in the PGCs or the immediate surrounding soma in E10.5 embryo (Fig 1 C and D). Since *Cdon* expression is clearly detectable in other areas, our negative results are not due to technical problems with the antibody or primers. For Reviewer's information, a zoomed-out image of the entire embryo cross-section after *Cdon* immunofluorescence (left) and *in situ* hybridisation (right) are shown below.

We noticed that the mouse strain used by Jameson et al., is in a CD1 background, while our mice are in a C57BL/6 background. CD1 and C57BL/6 mice can show significant differences in gene expression (as reported in Wang S et al (2019) *Toxicology* 412:19-28 and Zwemer CF et al (2007) *J Appl Physiol* 102:286-93) and indeed, some variabilities in the co-receptor mutant mice (as reported in Hong M et al (2013) *PLoS One* 8: e79269, Zhang W et al (2006) *Dev Cell* 10: 657-665, Cole F et al (2003) *Curr Biol* 13: 411-415, Zhang W et al. (2011) *Dis Model Mech* 4: 368-380 and Seppala M et al (2014) *Biol Open* 3: 728-740). Therefore, we speculate that the genetic background of mouse strains may underlie these discrepancies. We also noticed that the levels of *Gas1*, *Cdon* and *Boc* transcripts in NIH3T3 cells can be influenced by the presence or absence of serum in the culture medium (Supplementary Fig S5B). Therefore, we suspect that depending on the way the samples are prepared, mRNA levels of these genes could also be influenced.

2) I am not sure why you say 'PGCs exclusively express *Ptch2* and *Gas1* while ciliated somatic cells immediately surrounding PGCs express *Ptch1* and *Boc*' (line 129)? I think Figure 1D and Figure 5C suggest that *Boc* is also expressed by PGCs?

=> We agree that this sentence could be mis-interpreted. We have now modified the sentence to make it clearer (page 6, line 133).

3) Figure 5A – you isolate PGCs and somatic cells from GR by FACS. Why are these cells isolated from 'GR cultures' (not from freshly-dissected GRs?).

=> They were in fact single-cell suspensions isolated from freshly-dissected GR tissue homogenates. This sentence in the figure legend is now modified to make it clearer (page 27, line 659-60). We apologise for the error.

4) More importantly, I think that you could and should address cell type specificity of expression of *Gas1*, *Boc* (and *Cdon*) in these samples.

=> We now include the immunofluorescence staining results of all receptors of interest (*Ptch1*, *Ptch2*, *Gas1*, *Cdon* and *Boc*) in the primary GR cultures (Fig 5A). We performed additional qRT-PCR analyses for *Ptch1* and *Ptch2* on sorted cells (Fig 5B) because the expression of *Ptch1* is difficult to evaluate based on the immunofluorescence images alone, since they appear as small puncta.

5) I don't see any reason why the analysis of Gas1-null mice should be relegated to the Supplementary information.

=> The data are now included as Figure 10 in the main text.

6) The Ptch2 null mouse model is reportedly fertile (Lee et al, 2006 – your ref 34). This one seems a complete KO (while the one in ref 43 is a hypomorph). What does this mean in terms of Hh regulation of germ cell migration, particularly when considered in combination with the Gas1 null phenotype? Can you comment on this please in the Discussion.

=> We appreciate the very relevant comments by the reviewer. We recognize that in the original version of our MS these points were insufficiently discussed. We now include a paragraph discussing the potential implications of these findings in the Discussion (page 17, line 402-412)

7) The topic is complex but, nonetheless, the Introduction seems extremely long.

=> We have now shortened the Introduction by 281 words.

Minor points

8) Very weird to have a list of reference in the first paragraph as 23,31,49,52 (presumably these should be 1,2,3,4 and the Reference list amended accordingly).

=> We changed the referencing style and bibliography accordingly.

9) Line 63 – the word 'demonstrated' should probably be replaced by 'claimed' since the reports present conflicting views

=> We agree with the Reviewer and the word is now replaced (page 3, line 55).

10) In Figure 4c, the numbers 1 and 2 seem to be missing from the first panel

=> The missing numbers are added in Figure 4C. We apologise for the error.

11) Please note the timepoint at which PGCs were isolated from Stella-GFP embryos (line 218), Figure 5 legend, Supp Fig 2 legend. This appears only in the Methods (E10.5).

=> We have included the time points (E10.5) in the main text (page 9, line 202 & 205) and in the Figure 5B and Supplementary Fig S4 legends.

12) Saying the 'anlage of E9.5-11.5 embryos' doesn't really make sense (line 147). You should say 'gonadal anlage'?

=> The erroneously omitted word 'gonadal' is now added (page 6, line 117).

13) The videos are not labelled as S1, S2, S3, S4 – I assumed they are in numerical order (4033805, 6, 7, 8)

=> This must have been a technical glitch during online submission. We have re-labelled all video files (total 10 including the new motility assay data) accordingly.

In response to Reviewer #3:

Review report for manuscript NCOMMS-19-28270 entitled "Ptch2/Gas1 and Ptch1/Boc differentially regulate Hedgehog signalling in primordial germ cell migration"

In this manuscript, the authors report differential pathways of Hedgehog (HH) signaling mediated by receptor complexes, Ptch2/Gas1 and Ptch1/Boc. At the primordial germ cell (PGC)-migration stages

in mouse development (E10.5-E11.5), the authors report that Ptch2 and Gas1 are expressed only in PGCs, marked by Stella-EGFP or SSEA1, while Ptch1 and Boc are expressed in somatic cells. The HH signaling seems to enhance migratory activity of PGCs in embryonic slice cultures ex vivo, without affecting their directionality. Interestingly, Boc and Gas1 seems to be differentially required in the HH signaling pathways mediated by Ptch1 and Ptch2, respectively. The authors also report that PGCs do not have mature primary cilia, but PGCs activate Smo and Gli3, known downstream factors of the HH signaling associated with the cilia, in a different manner from the canonical pathway. Consistently, the authors describe that Gas1 mutant embryos show migratory defect of PGCs. Together, the concept of the alternative/non-canonical HH signaling specific to PGCs may be interesting. However, I have many significant concerns with on the data quality and data interpretations in this manuscript.

(1) Immunofluorescence analyses.

The quality of immunofluorescence (IF) is not very high. Even in the representative images, the fluorescence signals are often saturated perhaps due to too high signal gains (e.g., Fig 1D, SSEA1; Fig 5B DAPI). The IF images also often show too high background (e.g., Fig 5C DAPI). Such signal saturation and/or high background make it difficult to evaluate the IF data.

=> We appreciate the reviewer's cautiousness and have undertaken careful re-assessment of the images mentioned by the Reviewer. We can confirm that PGCs do express a very high level of SSEA, even when the image was taken at the lowest gain setting in the microscope. On the other hand, Boc shows a diffused weak signal which can be mistakenly viewed as a background. As a result, when these signals are overlaid with DAPI in the merged image, they can appear as if they are saturated or have a high background. Therefore, we now show the DAPI obtained at fixed gains as separate images to improve the clarity and visibility of individual signals.

Moreover, many IF panels show only one or two PGCs or somatic cells in the field of view with poor information of surrounding cells (e.g., Fig 4A and others). Because development proceeds with cellular interactions, such poor histological information also makes it difficult to evaluate the IF data.

=> We now include zoomed-out images of PGCs including more surrounding cells (Fig 5A). Please note, PGCs are sparsely localised as a single cell, scattering throughout a broad area, as shown in Fig 1D. Therefore, you rarely find multiple PGCs in one field of view. We analysed many independent images of PGCs and surrounding somatic cells to obtain consistent information. We also included a low magnification image as well as a zoomed-in image wherever possible.

In addition, this study often relies solely on visual inspection of representative images for the analyses of the IF data, and lacks statistical analyses.

=> We now include a "normalised", quantitative assessment of fluorescence images with appropriate statistical analyses based on multiple independent experiments. The graphs in Fig 4D, 5D and 9B show relative fluorescence intensity values from multiple cells which have been normalised to the fluorescence intensity of Stella or SSEA in each cell as internal reference value.

(2) Specific expression of Ptch2 and Gas1 in PGCs

The authors claim that Ptch2 and Gas1 are specifically expressed in PGCs but not in somatic cells (Fig 1D, 5A, 5B), and that they function in the PGC migration in a cell-autonomous manner. However, in Fig 1C, Gas1 is clearly expressed in somatic cells in hindgut, dorsal mesentery, and intermediate mesoderm at E9.5 and/or E10.5. All of these tissues are along the route of the PGC migration, and thus, Gas1 may function indirectly in the PGC migration.

=> At these stages, Hh signalling is active in this region, driving morphogenesis of the urogenital system as well as the neural tube. Therefore, the possibility that Gas1 mediates Hh signalling in different cell types cannot be completely ruled out. Moreover, Gas1 has been shown to be involved in other signal pathways (reference 53). Even in the case when endogenous Gas1 is present in some

cells, only a specific sub-population of these cells would co-express Ptch2. Therefore, the effects of Ptch2/Gas1-dependent Hh signalling in this region is likely to be limited.

=> To complement the data from embryos, we also demonstrate the motile behaviour (Fig 2D) and signalling responses of these cells using primary cultures (Fig 2D, Fig 4, Fig 9) where the individual PGCs and somatic cells are isolated from each other in single-cell suspensions and cultured as a mono-layer of cells. Therefore, it is most likely that they were cell-autonomous direct effects.

In addition, in Fig 5A and 5B, two different methods are used to evaluate the expression of Ptch1/Ptch2 and Gas1/Boc; Ptch1 and Ptch2 are analyzed using real-time PCR (Fig 5A) while Gas1 and Boc are analyzed using IF (Fig 5B). This is strange to me. Why do not the authors use the same method consistently for these factors? (i.e., IF for both or real-time PCR for both).

=> We now include the immunofluorescence analyses of all factors (Ptch1, Ptch2, Gas1, Boc and Cdon) in the revised Fig 5A. We performed additional qRT-PCR analyses only for *Ptch1* and *Ptch2* in sorted cells to further validate the differential expression of *Ptch1* which appears as small puncta in the immunofluorescence, thus difficult to evaluate based on the images alone.

Moreover, for the real-time PCR analysis, according to Materials and Methods, $2^{\Delta\Delta Ct}$ values are plotted and tested in Fig 5A. However, because the raw data of real-time PCR are Ct values, statistical tests should be also performed for Ct values or their linear derivatives (e.g., delta-delta Ct values).

=> We include the delta-delta Ct values and the Ct values with statistical analysis as below. The raw data are also provided in the Source Data Files.

Delta calculation

GFP	Population	18s Ct	GFP Ct	Δ Ct	$\Delta\Delta$ Ct	Average $\Delta\Delta$ Ct	S.D. $\Delta\Delta$ Ct	
PGCs	GFP+	20.54	25.73	5.19	-4.05	-4.06	0.0361	
	GFP+	20.54	25.68	5.14	-4.10			
	GFP+	20.54	25.75	5.21	-4.03			
Soma	GFP-	16.46	25.66	9.20	-0.04	-0.00333	0.0473	
	GFP-	16.46	25.68	9.22	-0.02			
	GFP-	16.46	25.75	9.29	0.05			
Control average Δ Ct				9.24				unpaired t-test p<0.0001
Stella	Population	18s Ct	Stella Ct	Δ Ct	$\Delta\Delta$ Ct	Average $\Delta\Delta$ Ct	S.D. $\Delta\Delta$ Ct	
PGCs	GFP+	20.54	16.48	-4.06	-4.22	-4.27	0.0987	
	GFP+	20.54	16.32	-4.22	-4.38			
	GFP+	20.54	16.50	-4.04	-4.20			
Soma	GFP-	16.46	16.65	0.19	0.03	0.00	0.1082	
	GFP-	16.46	16.50	0.04	-0.12			
	GFP-	16.46	16.71	0.25	0.09			
Control average Δ Ct				0.16				unpaired t-test p<0.0001
Ptch1	Population	18s Ct	Ptch1 Ct	Δ Ct	$\Delta\Delta$ Ct	Average $\Delta\Delta$ Ct	S.D. $\Delta\Delta$ Ct	
PGCs	GFP+	20.54	37.98	17.44	1.04	1.03	0.0153	
	GFP+	20.54	37.99	17.45	1.05			
	GFP+	20.54	37.96	17.42	1.02			
Soma	GFP-	16.46	32.86	16.40	0.00	0.00333	0.1550	
	GFP-	16.46	33.02	16.56	0.16			
	GFP-	16.46	32.71	16.25	-0.15			
Control average Δ Ct				16.40				unpaired t-test p=0.0070
Ptch2	Population	18s Ct	Ptch2 Ct	Δ Ct	$\Delta\Delta$ Ct	Average $\Delta\Delta$ Ct	S.D. $\Delta\Delta$ Ct	
PGCs	GFP+	20.54	33.20	12.66	-7.23	-6.86	0.3512	
	GFP+	20.54	33.60	13.06	-6.83			
	GFP+	20.54	33.90	13.36	-6.53			
Soma	GFP-	16.46	36.22	19.76	-0.13	0.00	0.1179	
	GFP-	16.46	36.45	19.99	0.10			
	GFP-	16.46	36.38	19.92	0.03			
Control average Δ Ct				19.89				unpaired t-test p=0.0013

(3) Migration assay of PGCs in the slice culture

The authors assay migratory activity of PGCs using embryonic slice cultures exposed to agonist and antagonist of HH (Fig 2 and 3). By tracing the stella-EGFP-positive PGCs, the authors conclude that the HH signaling enhances the PGC migration in a non-directional manner. I failed to find the information how the authors chose the concentrations of the agonist/antagonist (40uM).

=> We were well aware of the potential toxicity, especially with cyclopamine, therefore we screened various concentrations of each drug in our embryo slice culture condition. We did not see any effects of cyclopamine at concentrations up to 20 μM but started seeing a reduction in PGC motility at 40-60 μM without any sign of general cytotoxicity. We only noticed some slight toxicity when 80 μM was applied. This was the highest concentration we tested, at which the solvent control also showed slight toxicity. Similarly, we tested varying concentrations of purmorphamine and found 40-60 μM was showing effects on motility with no toxicity. We now include these optimisation data in Supplementary Figure S2 A-C. Based on these studies, we used the drugs at 40 μM final concentration. We also examined the effects of the drugs on overall survival of the PGCs (GFP⁺) by assessing the number of hours that the green fluorescence signal could be observed in the live-imaging (Fig 3C), and found no difference among treatment groups.

Moreover, the embryonic slices cultured with the agonist/antagonist rapidly proliferate or grow with very different kinetics accompanying macroscopic morphological changes as shown in the supplementary videos. Thus, the agonist/antagonist clearly act on the surrounding somatic cells, and the change of the migration speeds of PGCs may be an indirect and/or apparent effect.

=> To quantitatively evaluate the growth rate of the embryos over the experimental time frame in the supplementary videos, we measured the total area of the embryo trunk at the beginning and end of the movie (t=0 and t=10hr) as shown in Supplementary Figure S3. The data confirmed that the ratio of embryo size expansion was similar in all treatment groups, with approximately 2-fold increase over the same time interval (Fig 3D). Therefore, the changes in the PGC migration speed were not indirect or apparent effects caused by the morphological changes of the surrounding somatic tissues. The data are now discussed in the main text (page 8, line 166-167).

In addition, if the HH signaling enhances the PGC migration, it would be expected that PGCs may form more lamellipodia and/or filopodia. Thus, cellular morphology of PGCs under influence of the HH signaling should be investigated.

=> It has been documented that PGCs migrate via the “bleb-associated amoeboid” mode (Blaser et al. Dev Cell 11:613–627. Also reviewed in JCB 181:879–884). Meaningful analysis of blebbing movement requires specialised evaluation techniques including 3D cell volume assessment, internal mobilization of cytoplasm and the hydrodynamics motion analysis under high speed live-imaging camera at a high magnification, in addition to the classical F-actin staining for lamellipodia and filopodia, which will be beyond the scope of this paper, but we hope to investigate these aspects as an independent project.

(4) Fig 4A: Cytoplasm localization of Smo and Gli3

The authors report that the protein levels of Smo and Gli3 increase in cytoplasm when PGCs are treated with a HH agonist (Page 9, Line 207) (Fig 4A). However, Fig 4A clearly shows that signals of Smo and Gli3 are overlapped with the DAPI signal, contradicting the authors' claim.

=> PGCs have very big nuclei with relatively small cytoplasm. The merged images shown in the original Fig 4A had been also overlaid with Stella (cytoplasmic signal) which could be mistakenly viewed as a nuclear signal when overlaid with DAPI. For greater clarity, we have now removed the Stella signal in the merged images so that the extra-nuclear cytoplasmic localisation of Smo and Gli3 in relation to DAPI are more visibly distinguished.

(5) Fig 4B: Quantification of IF intensities

The IF intensities of Smo and Gli3 are quantified and compared between PGCs treated with purmorphamine (PUR) and DFM (DMF (dimethyl formamide?)) (Fig 4B). However, quantitative measurement of fluorescence intensity of IF is generally difficult and needs careful interpretation. For example, in order to control the variation of staining, the two cell types of which IF intensities

are compared (i.e., +DFM and +PUR cells) should be intermingled on the same slide glasses and stained at the same time.

=> We appreciate the Reviewer's cautiousness in this type of analyses. In our experiments, cells in different treatment groups were in fact plated on a multi-chamber glass slide, which were processed together. The identity of sample was randomised to the researcher and images were taken blindly using the same microscopic setting. To consider the staining variation between cells, we have used the fluorescence intensity of SSEA as the internal reference value in each cell and normalised the fluorescence intensity of Smo and Gli3 against it. We now include a revised graph using these "normalised" values (Fig 4B and Fig 5D).

=> We thank the Reviewer for pointing out the typo of DMF, which is now corrected throughout.

(6) Fig 4C: un-ciliated PGCs

The authors report that PGCs are un-ciliated based on the lack of the dots of Arl13b and Ac-T, markers for primary cilia (Fig 4C). However, Fig 4C also shows that there are somatic cells negative for the dots of the cilia markers. Thus, it is required to show statistics of PGCs and somatic cells that are positive/negative for these markers.

=> We thank the Reviewer for this helpful suggestion. We now include a data graph showing the percentage of ciliated population in PGCs versus somatic cells with statistical analyses (Fig 4D).

(7) Fig 5C: "E13.5 PGCs".

The authors analyze the p-Src level in "E13.5 PGCs" to evaluate the Sonic Hedgehog (Shh) signaling mediated by the Ptch2/Gas1 complex in non-migratory germ cells (Fig 5C). However, I failed to identify the sex of these germ cells: At E13.5, mouse gonads clearly show a morphological difference between male and female, and germ cells also undergo clear sex differentiation (mitotic arrest in male, and meiotic entry in female). Moreover, because E13.5 germ cells are at quite distinct cell-cycle phases (mitotic arrest or meiosis) from the E10.5 PGCs (mitotically active), I think that it may not be appropriate to argue that Ptch2/Gas1 is functionally important for migration because p-Src is positive in E10.5 PGCs and is negative in E13.5 germ cells.

=> We appreciate the Reviewer's point that E13.5 is not the appropriate stage to investigate PGC motility, due to the various ongoing changes such as sex differentiation and gonad morphogenesis. We realised that a better argument can be made by comparing the status of PGCs in a Gas1-deficient context. Therefore, we now include data from *Gas1*^{-/-} embryos at E10.5 (Fig 5C & D). In comparison to the WT, Gas1 KO PGCs show a reduced level of p-Src, potentially underlying the delayed migration and ectopic PGCs present in Gas1 KO embryos. These data support the functionally important role of Gas1 in Ptch2-dependent signalling that regulate PGC motility.

(8) Fig 6 and 7: Over-expression of Ptch1 and Ptch2 in the Ptch1-KO NIH3T3 cells

Using Ptch1/Ptch2-over expressing cells, the authors perform Co-IP analyses for the Ptch1/Ptch2 proteins to assay the binding activities of Gas1, Boc, and Shh to these receptors (Fig 6A, 6C and 7A). The authors also assay the levels of downstream factors, pCreb, pSrc, and the Gli3 cleavage (Fig 6E). However, I failed to find the methods and primary results of the over-expression. Which promoters are used, and how? Are Ptch1/Ptch2 expressed transiently or stably? What are the expression levels of the over-expressed proteins?

=> We apologise for the omission of detailed information regarding the Ptch1/Ptch2 expression vectors, which is now included in the revised Materials and Methods (page 22, line 518-519). The transiently transfected, CMV promoter-driven Ptch1 and Ptch2 constructs reproduced respective proteins at comparable levels as shown in Fig 6A, 6C and 7A.

Are the over-expression levels comparable to endogenous/physiological levels in NIH3T3 cells and/or PGCs? Are the over-expression levels of Ptch1 and Ptch2 similar to each other sufficiently for

the comparison of the signal transduction activities? All of the above are critical for the appropriate evaluation of the experiments described in this manuscript.

=> Due to the delicate nature of the PGCs which only exist in small numbers in early stage embryos, we employed NIH3T3 cells to test our hypothesis at the molecular level. Comparison of exogenously transfected *Ptch1/2* expression in NIH3T3 to the physiological levels of proteins in PGCs is difficult, if not impossible, as they are very different types of cells. Despite the limitations of *in vitro* studies, the NIH3T3 model allows us to introduce only one receptor at a time in a *Ptch1/Ptch2*-null background so that we can study the receptor-specific mode of action without interference from the others – otherwise, *Ptch2*-dependent activities are normally obscured by the effects of *Ptch1*, as previously reported. Our reverse Co-IP experiments using endogenous *Gas1/Boc/Cdon/Smo* proteins as the bait have brought down a similar level of the transfected *Ptch1/2* proteins, indicating that the stoichiometric interactions among these proteins were comparable in the unstimulated state. Previous publications, which originally described the binding interactions of these co-receptors with *Ptch1* and *Shh* ligand had extensively relied on the overexpression approaches (i.e. co-transfections of multiple constructs) and did not examine the time-dependent interactions with endogenous binding partners as demonstrated by our current study.

(9) Fig 6E: Gli3 FL/R ratio.

The authors report that *Ptch1* and *Ptch2* show different kinetics of stabilization of Gli3 protein in response to *Shh* signaling based on the ratio of full-length and cleaved Gli3 (Gli3 FL and Gli3 R, respectively) on the Western blot analyses (Fig 6E). However, the total protein level of Gli3 seems to change significantly upon the *Shh* treatment; e.g., in the *Ptch2* over-expressing cells, the Gli3 level decreases transiently at 40 min after the *Shh* treatment and then drastically increases at 24 hours. With such a change of the protein level, it may not be appropriate to simply interpret the Gli3 FL/R ratio as an indicator of the Gli3 stabilization level. Moreover, the quantification of the Gli3 FL/R ratio should be repeated for several times with a statistical test. In addition, the logic of the interpretation of the Gli3 FL/R ratio is not clear to me: this ratio changes from 0.3 to 0.6 (twice) in the *Ptch2* over-expressing cells at 40 min, while it changes from 0.2 to 0.7 (3.5 times) in the *Ptch1* over-expressing cells at the same time (Fig 6E, left panels). With a hypothesis that this ratio indicates the Gli3 stabilization level (i.e., HH signaling level), the authors claim that *Ptch2* transduces the HH signaling rapidly, while *Ptch1* transduces it gradually. However, simply based on the Gli3 FL/R ratio above, the following interpretation is also possible; Gli3 is more rapidly stabilized in the *Ptch1* over-expressing cells, and this may lead us to speculate that *Ptch1* transduces the HH signaling more rapidly, and the signal transduction continues to be reinforced for at least 24 hrs, even after *Boc* is released from *Ptch1* (Fig 7A), while *Ptch2* transduces it more slowly and less efficiently thereby reaching the plateaux at a low level at short times.

=> We appreciate the Reviewer's insightful suggestion. To obtain a more quantitative understanding, we conducted qRT-PCR analyses of *Gli1* and *Gli3* mRNA after *Shh* treatment, a classical approach used to investigate Hh signalling activity. Our new data from 3 biological replicates with statistical analyses in each cell line (Fig 8A) demonstrate that *Ptch1* and *Ptch2* show a different signalling profile over the key time points. While *Ptch1*-dependent Hh signalling gradually induces *Gli1* and *Gli3*, reaching a maximum at 24 hours; *Ptch2*-dependent Hh signalling induces them only to a moderate level, even at 24 hours. Therefore, *Ptch2* could mediate canonical Hh signalling but rather inefficiently. Nonetheless, *Gas1* KO completely abolished *Ptch2*-dependent responses, but not those of *Ptch1*-dependent, confirming the specific obligatory requirement for *Gas1* in *Ptch2* function. A paragraph describing the new data is now included in the main text (page 12, line 272 - 278).

(10) PGCs in the *Gas1*-knockout mice

The authors report a migratory delay and existence of ectopic PGCs in the *Gas1*-mutant embryos (Fig S5A, S5B). However, *Gas1* deficiency may affect PGC specification or survival, and thus, total number of PGCs in the mutant embryos should be counted.

=> The data in our original manuscript was based on total PGC counts. Paraffin-embedded embryos were sectioned in 5µm and SSEA-positive PGCs counted from every other section of the entire embryo. The total number of PGCs in the genital ridge, mesentery and hindgut was counted per embryo from each genotype. We are sorry that this information was not clearly provided, and now include a more detailed description in the figure legend (Fig 10B).

=> Since *Gas1* KO PGCs still expressed germ-cell specific marker genes, such as SSEA and Stella; *Gas1* KO did not affect the fate determination or specification of PGCs. Moreover, despite the presence of ectopic PGCs in the *Gas1* KO embryos indicating delayed migration, the total number of PGCs (combined from all regions) was not significantly reduced, therefore, overall survival of PGCs was not affected by *Gas1* KO compared to WT. We have now modified the graph to show the total number of PGCs in each genotype (Fig 10B) and discussed this point in the main text (page 14, line 337-339).

Reviewers' Comments:

Reviewer #1:

Remarks to the Author:

To an extent my concerns have been addressed although and I will take the high concentration of the small molecules at face value. It is of note that vismodegib and cyclopamine have been tested on explants and had effects at much lower concentrations

Point 7 (Figure 8): The authors adhere to the canon that Ptch1 or 2 activity inhibits Smo, and released upon Shh binding in conjunction with a co-receptor. They confirm earlier results that Ptch2 can mediate the transcriptional Shh response in the absence of Ptch1. They show that overexpression of in particular Ptch1, but to a lesser extent Ptch2 can restore the Hh response as measure by Gli expression. As this induction would entail a de-repression of Smo, it follows that in transfected cells the Hh response should be high, as they lack Ptch1/2 inhibition altogether. A direct comparison of the mock, Ptch1 and Ptch2 transfected cells is indicated, as according to the model, the Ptch1 expressing Shh-induced cells should have a similar level of Gli expression as the mock transfected cells. This is not apparent from the data as presented as they are normalized to the untreated condition, which confusingly is different for the 0 timepoints. Regardless, this figure does provide confirmation that the relative response to Shh of overexpressed Ptch2 and Ptch1 are distinct, but at least presented in a way that Ptch2 and Ptch2 have activating roles rather than de-repressing roles. At least worth discussing is the observations that cells lacking Ptch1/2 remain Shh sensitive and Smo-dependent in a Boyden chamber assays, and the absence of the 24C palmitoylation, that is both required for Ptch1 regulation and has a prominent location in the Ptch1/Shh structures.

I remain somewhat skeptical about the Co-IPs. For sure they all involve Ptch2 overexpression and are thus susceptible to associated artifacts. I would suggest to move the requested no-bait control to the main figure, there is plenty of space.

The Smo IP by Ptch2 looks OK. I wonder why the direct SmoIP looks so poor, as this should more easily detect Smo. In addition, less Smo is detected in the +Shh condition, undermining the results of the Ptch2 co-IP. The direct Smo WB looks confusing too, and I wonder which of the several bands is the form of Smo of interest.

Reviewer #2:

Remarks to the Author:

The authors have undertaken extensive review and modification of the manuscript and figures. This has included new experimental work, and the inclusion of additional supplementary data. This has greatly improved the manuscript. The serious reservations expressed by reviewers have been dealt with, in my opinion.

Reviewer #3:

Remarks to the Author:

Review report for the revised version of manuscript NCOMMS-19-28270A.

I have reviewed the revised version of the manuscript, and found that the authors extensively addressed all of my concerns.

Now I have only one point to suggest further: In Figure 4A, DAPI may be shown in separate panels so that the cytoplasmic localization of Smo and Gli3 becomes clearer.

The authors would be willing to address the above point, thereby I can recommend this study for

publication.

We respond to each of the reviewers' specific points as below.
The changes are also highlighted in the main text.

In response to Reviewer #1:

To an extent my concerns have been addressed although and I will take the high concentration of the small molecules at face value. It is of note that vismodegib and cyclopamine have been tested on explants and had effects at much lower concentrations

=> We thank the reviewer for the helpful suggestion. We have now included a sentence discussing this in the main text (page 8, line 178-179).

Point 7 (Figure 8): The authors adhere to the canon that Ptch1 or 2 activity inhibits Smo, and released upon Shh binding in conjunction with a co-receptor. They confirm earlier results that Ptch2 can mediate the transcriptional Shh response in the absence of Ptch1. They show that overexpression of in particular Ptch1, but to a lesser extent Ptch2 can restore the Hh response as measure by Gli expression. As this induction would entail a de-repression of Smo, it follows that in transfected cells the Hh response should be high, as they lack Ptch1/2 inhibition altogether.

=> We agree with the Reviewer's speculation that baseline activity of the pathway must be higher due to the lack of repression activities of Ptch. However, our empty vector transfected *Ptch1* KO cells (i.e. *Ptch1/2* double null) do not show a uniformly high level of constitutive signalling activity, which agrees with previously published data by other independent groups showing that an absence of Ptch1/2 does not lead to full activation of the pathway. This might be due to the fact that without Ptch receptors, cells are not able to receive the ligand properly, thus the intrinsic sensitivity to Hh may become impaired or even desensitised. In the case that the baseline activity levels become elevated in our double null cells, it is common to all cells we analyse in this experiment, thus represents the background. The key finding we demonstrate here is, Ptch1- or Ptch2-expressing cells respond to the ligand treatment in a time and dose-dependent manner above background. And this "ligand-induced response" is differentially affected by the targeted removal of Gas1.

A direct comparison of the mock, Ptch1 and Ptch2 transfected cells is indicated, as according to the model, the Ptch1 expressing Shh-induced cells should have a similar level of Gli expression as the mock transfected cells. This is not apparent from the data as presented as they are normalized to the untreated condition, which confusingly is different for the 0 timepoints.

=> "Ptch1 expressing Shh-induced cells" do not have a similar level of Gli expression as the mock transfected cells because the "mock transfected cells" are *Ptch1* KO cells without any Ptch1/2 function (i.e. double null). Therefore, the mock transfected cells won't respond to Shh ligand and cannot activate the downstream canonical signalling. This is clearly shown in our source data files where the un-normalised values at 0 time-points are different between the empty vector and the Ptch1 transfected cells. We observed that the introduction of Ptch1 makes the cells regain sensitivity to the endogenous Hh ligand, resulting in a slight increase of basal levels, which gets further increased upon Shh-N treatment at 40 minutes and 24 hours.

Regardless, this figure does provide confirmation that the relative response to Shh of overexpressed Ptch2 and Ptch1 are distinct, but at least presented in a way that Ptch2 and Ptch1 have activating roles rather than de-repressing roles.

=> We found that *Ptch1/2* double null cells do not thrive in general and grow rather slowly. Notably, functional interactions between autophagy, ciliogenesis and Hh signalling has been reported (*Pampliega et al (2013) Nature 502:194-200*). Therefore, it is likely that basal level Hh signalling is required for the maintenance and survival of cells. As the Reviewer rightly points out, re-introduction of either of the receptors reinstated the cells' sensitivity to Hh and increased the basal

level activity even at 0 time-points. From our point of view, whether this can be interpreted as having an “activating” role is a matter of debate.

At least worth discussing is the observations that cells lacking Ptch1/2 remain Shh sensitive and Smo-dependent in a Boyden chamber assays, and the absence of the 24C palmitoylation, that is both required for Ptch1 regulation and has a prominent location in the Ptch1/Shh structures.

=> We thank the reviewer for the helpful suggestion. We have now included a sentence discussing this in the main text (page 17-18, line 414-416).

I remain somewhat skeptical about the Co-IPs. For sure they all involve Ptch2 overexpression and are thus susceptible to associated artifacts. I would suggest to move the requested no-bait control to the main figure, there is plenty of space.

=> We have now moved the no-bait control experiment into Figure 7 as requested.

The Smo IP by Ptch2 looks OK. I wonder why the direct SmoIP looks so poor, as this should more easily detect Smo. In addition, less Smo is detected in the +Shh condition, undermining the results of the Ptch2 co-IP. The direct Smo WB looks confusing too, and I wonder which of the several bands is the form of Smo of interest.

=> Smo undergoes various post-translational modifications including N-glycosylation (*Marada et al, PLoS Genet 11(8): e1005473*) and serine/threonine phosphorylation at multiple sites (*Chen & Jiang (2013) Cell Res 23:186–200*), which causes Smo to appear in multiple bands in Western blot, ranging from about 80kDa to 150kDa. The significance of these modifications in the regulation of Smo activity is not fully understood, but according to a study (*our reference 26*), the two predominant bands with apparent molecular weight of 110kD (~100kD in our gel) and 95kD (~85kD in our gel) are likely produced by differential glycosylation, as the 95kD product is sensitive to endoglycosidase H (Endo H) treatment, while the 110kD product is more resistant. Therefore, the 110kD band represents the more extensively glycosylated form of Smo. These forms of Smo are suggested to be differentially localised at the plasma membrane upon Shh stimulation, affecting the trafficking and activity of Ptch/Smo. It is well established that the Ptch/Smo complex traffics to the endosome after ligand binding and Smo is subsequently segregated from Ptch1 and destined for lysosomal degradation. This is probably why less Smo is detected in the +Shh condition in our gel.

=> For Reviewer’s information, we include the image captures of Smo Co-IP western blot data from a few independent publications as below.

COS-7 cells co-transfected with Flag-Ptch1 or Flag-Ptch2 and Myc-Smo. IP with anti-Flag and Western blotting with anti-Myc.
Figure 2 from Carpenter et al (1998) PNAS 95:13630-13634

COS cells co-transfected with Ptch1-GFP and Smo-Flag. Lane 1: IP with anti-Smo and Western blotting for Ptch1. Lane 3: IP with anti-Ptch1 and Western blotting for Smo. Lanes 2 and 4 are negative controls using preimmune serum.
Figure 3 from Karpen et al. (2001) J. Biol. Chem. 276:19503-19511

HEK cells transiently transfected with Myc-Smo, Flag-Gas8 or both. IP with anti-Flag and Western blotting for Myc or Flag. Whole cell lysate (WCL) western blot for Myc-Smo.
Figure 1 from Evron et al. (2011) J. Biol. Chem. 286:27676-27686

HEK cells transiently transfected with GFP-Smo and Flag-Invs. IP with anti-GFP and Western blotting for FLAG or GFP.
Figure 6 from Zhang et al. (2019) PNAS 116:874-879.

**Please note the size of Smo is about 30kD bigger due to the GFP-tag.*

In response to Reviewer #2:

The authors have undertaken extensive review and modification of the manuscript and figures. This has included new experimental work, and the inclusion of additional supplementary data. This has greatly improved the manuscript. The serious reservations expressed by reviewers have been dealt with, in my opinion.

=> We thank the reviewer for the positive comments. There is nothing to respond.

In response to Reviewer #3:

I have reviewed the revised version of the manuscript, and found that the authors extensively addressed all of my concerns.

Now I have only one point to suggest further: In Figure 4A, DAPI may be shown in separate panels so that the cytoplasmic localization of Smo and Gli3 becomes clearer.

The authors would be willing to address the above point, thereby I can recommend this study for publication.

=> We thank the reviewer for the positive comments. We have made the changes in Figure 4A as suggested.